# Isolable f-element diphosphene complexes by phosphinidene group transfer and coupling at uranium

Jingzhen Du [1,3], Thayalan Rajeshkumar [2], John A. Seed[1], Ashley J. Wooles [1], Laurent Maron [2] ✉ & Stephen T. Liddle [1] ✉

The parent diphosphene (HPPH) molecule is of fundamental interest, but its reactive nature renders it challenging to isolate and study. Metal-stabilization is an attractive approach for studying HPPH, but molecular derivatives are limited to three complexes of p-/d-metals reflecting a scarcity of synthetic methods for rationally preparing HPPH complexes. Here, we introduce f-element HPPH complexes, adding to f-element diazenes (HNNH) that were first reported over thirty years ago. By utilizing $7\lambda^3$-phosphadibenzonorbornadiene and uranium(III) reagents we show how parent diphosphene, phosphinidiide, and diphosphorus motifs can all be constructed, developing synthetic approaches for this area. Computed reaction profiles reveal common, initial reaction steps that subsequently diverge depending on the ancillary ligands, radical nature of intermediates, and the $7\lambda^3$-phosphadibenzonorbornadiene P-substituent. Calculations demonstrate a surprising prevalence of open-shell radical intermediates, and that the redox chemistry is P-, not U-, centred. This work thus provides insights to inform future synthetic endeavours in this area.

As the first heavier Group 15 analogue of diazene, HNNH, the parent diphosphene, HPPH, represents a fundamentally important class of species in phosphorus chemistry[1–6]. However, HPPH possesses a relatively weak $P=P$ double bond and spatially expansive lone pairs that render HPPH highly reactive and so intrinsically challenging to isolate and study. Although HPPH was proposed in a mass spectrometry study of the decomposition products of $PH_3$ in 1966[7], spectroscopic identification of HPPH and its isomer diphosphinyldene $PPH_2$ was only achieved in 2023[8], and thus investigations of HPPH have largely remained restricted to computational analyses[9–17]. In contrast to the free form of HPPH, stabilization of HPPH in the condensed phase by metal centres is in principle a practical way to isolate and study this fundamental species[18–21]. However, molecular complexes containing HPPH remain exceedingly rare, being limited to two d-block complexes, $[(\eta^5\text{-}C_5H_5)_2Mo\{\mu\text{-}\eta^2{:}\eta^2\text{-}(HPPH)\}]$[18,19] and $[(\eta^5\text{-}C_5H_5)_2Ta(H)\{\mu\text{-}\eta^2{:}\eta^2\text{-}(HPPH)\}]$[20] prepared from $P_4$ and metal hydride precursors, and one p-block complex $[\{LGe\}_2\{\mu\text{-}\eta^1{:}\eta^1\text{-}(HPPH)\}]$ $(L = CH\{(CMe)(2,6\text{-}^iPr_2C_6H_3N)\}_2)$[21] isolated from a phosphanide complex by dehydrocoupling. Thus, to date, a HPPH complex remains elusive in f-element chemistry even though HNNH[22,23] and HAsAsH[24] f-element derivatives, prepared by acid-base methodologies, were reported in 1992 and 2015, respectively, Fig. 1. Therefore, fundamental questions remain over how HPPH might bind to f-element centres, for example as the neutral diphosphene $((HPPH)^0)$ or as the diphosphane-1,2-diide form $((HPPH)^{2-})$. The paucity of isolable complexes in this area reflects the intrinsic challenge of pairing the soft HPPH ligand to hard f-element metal ions from Hard-Soft Acid-Base (HSAB) arguments, and also a developing but still limited range of reliable and controllable synthetic methodologies for constructing novel heavier Group 15 motifs at actinide centres generally[25–36].

[1]Department of Chemistry and Centre for Radiochemistry Research, The University of Manchester, Manchester, UK. [2]LPCNO, CNRS & INSA, Université Paul Sabatier, Toulouse, France. [3]Present address: College of Chemistry, Zhengzhou University, Zhengzhou, China. ✉e-mail: laurent.maron@irsamc.ups-tlse.fr; steve.liddle@manchester.ac.uk

In preliminary work[37], we demonstrated that the $7\lambda^3$-phosphadibenzonorbornadiene (PDBN) compounds $Me_2N$-PDBN (**1**)[38] and H-PDBN (**2**)[39] are excellent reagents for generating diphosphorus, $P_2^{2-}$, and phosphinidiide, $HP^{2-}$, functional groups when reacted with low-valent $[(Tren^{TIPS})U^{III}]$ (**I**, $Tren^{TIPS} = \{N(CH_2CH_2SiPr^i_3)_3\}^{3-}$), Fig. 2[40]; this generated the f-element $P_2$ complex $[\{(Tren^{TIPS})U^{IV}\}_2(\mu-\eta^2:\eta^2-P_2)]$ (**II**), closing the gap generated by the discovery of $N_2$ and $As_2$ analogues in 1988 and 2015, respectively[24,41], Fig. 1, and a rare diuranium(IV)-phosphinidiide complex $[\{(Tren^{TIPS})U^{IV}\}_2(\mu-PH)]$ (**III**)[37], respectively, where in each case the P-moieties carry a 2− charge[37]. Preliminary reactivity studies revealed that **II** can be converted to cyclo-$P_3^{3-}$ or $P_2^{3-\bullet}$ derivatives in the presence of strong reductants[37,42], and **III** can be deprotonated to access a bridging phosphide[29]. Thus, **II** and **III** can be elaborated emphasizing the desirability of developing synthetic methodology in this area[43–46]. The formation of **III** and **II** suggests that,

formally, HP and $Me_2NP$ phosphinidene group transfers occurred, and for the former the putative $U^V = PH$ intermediate undergoes comproportionation with trivalent **I** to give tetravalent **III** whereas for the latter reductive P-N cleavage, experimentally confirmed by the isolation of $[(Tren^{TIPS})U^{IV}(NMe_2)]$ (**IV**)[37], and P-P coupling produces **II**. Given that **II** forms, this prompts the question of why **III** forms at all instead of the putative U = PH intermediate simply coupling to give a HPPH derivative. Evidently, the P-H bond in these reactions is more robust than the P-N bond, and analogously to the formation of **IV** the corresponding U-H bond from P-H cleavage would not be anticipated to be as favourable. However, it can be postulated that sterics will play a significant role, and presumably the HPPH unit is large enough to resist formation between two sterically demanding $Tren^{TIPS}U$ fragments. It follows that control over the reaction products will rest on the balance between reaction rates, phosphinidene group transfer, P-R cleavage, and steric control over any P-P catenation steps. To test this conjecture, we examined the reactivity of **1** and **2** with two $U^{III}$ complexes with sterically varied supporting ligands, namely $[(Tren^{DMBS})U^{III}]$ (**3**, $Tren^{DMBS} = \{N(CH_2CH_2NSiMe_2Bu^t)_3\}^{3-}$)[47] and $[(Ar*O)_3U^{III}]$ (**5**, $Ar*O = 2,6$-$Bu^t_2C_6H_3O^-$)[48], respectively.

Here, we introduce the diphosphene ligand to f-element chemistry through the synthesis of two U-diphosphene complexes, and also report the isolation of new diphosphorus and phosphinidiide U-complexes. By identifying divergent and parallel reactivity patterns, which suggest some universality in terms of underlying P-P coupling methodology, we elucidate the factors that direct diphosphene, diphosphorus, or phosphinidene formation. These insights offer potential to guide, and render more rational, future synthetic endeavours in this area, generally that complement methodologies for N- and As-analogues. Quantum chemical calculations show that the coordinated HPPH units are present in their phosphane-1,2-diide forms, consistent with prior predictions of the strong π-acceptor nature of HPPH. Lastly, quantum chemical reaction profile calculations unexpectedly suggest the presence of open-shell P-radical

**Fig. 1 | Preeminent examples that introduced $M_2E_2$ and $M_2E_2H_2$ structural motifs to f-element chemistry (M = Sm, U; E = N, P, As) in the given years.** The illustrations are simplified, with [] indicating the presence of M co-ligands; see references [22–24,36,40], and this study for details.

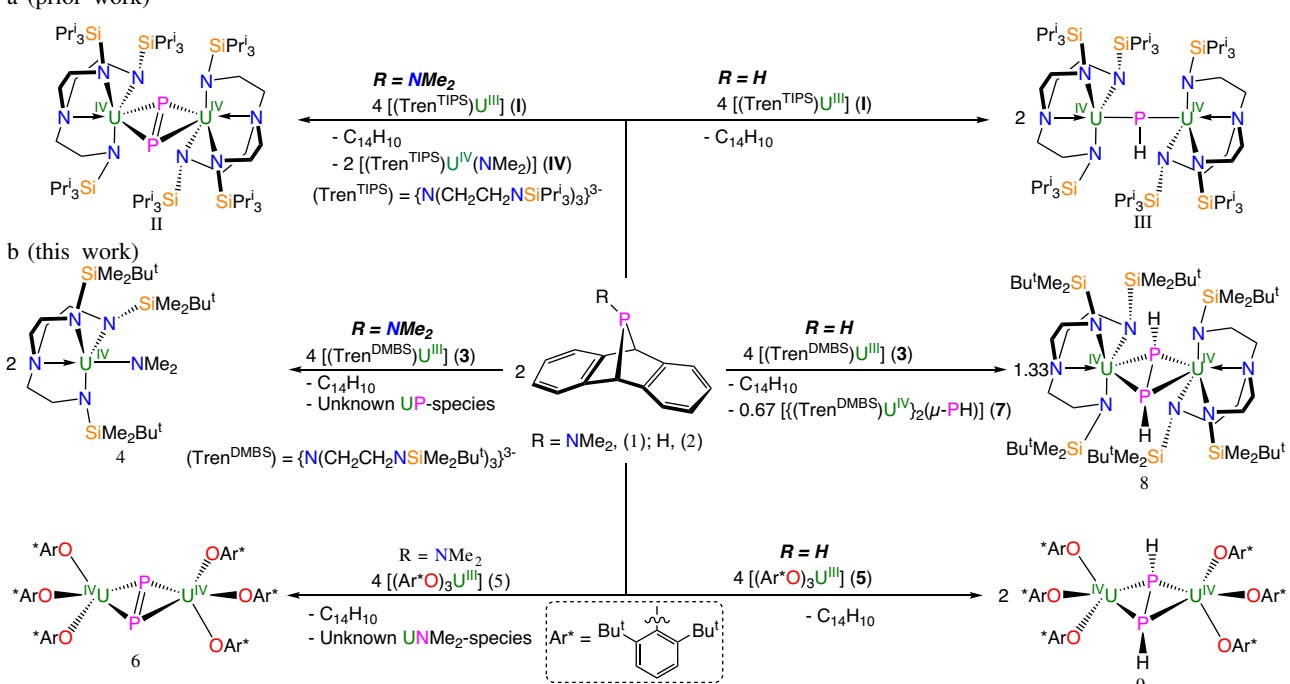

**Fig. 2 | Reactions of R-PDBN reagents with uranium complexes. a** prior work, see ref. 36, showing the reaction of **I** with **1** to give the diphosphorus complex **II** and with **2** to give the phosphinidene complex **III**. **b** this work showing the reaction of **1** with **3** and **5** to give the amide complex **4** and the diphosphorus complex **6**, with

unidentified byproducts, respectively and reactions of **2** with **3** and **5** to give the diphosphene complexes **8** and **9** (shown to be the diphosphane-1,2-diide forms) along with the minor phosphinidiide complex **7**.

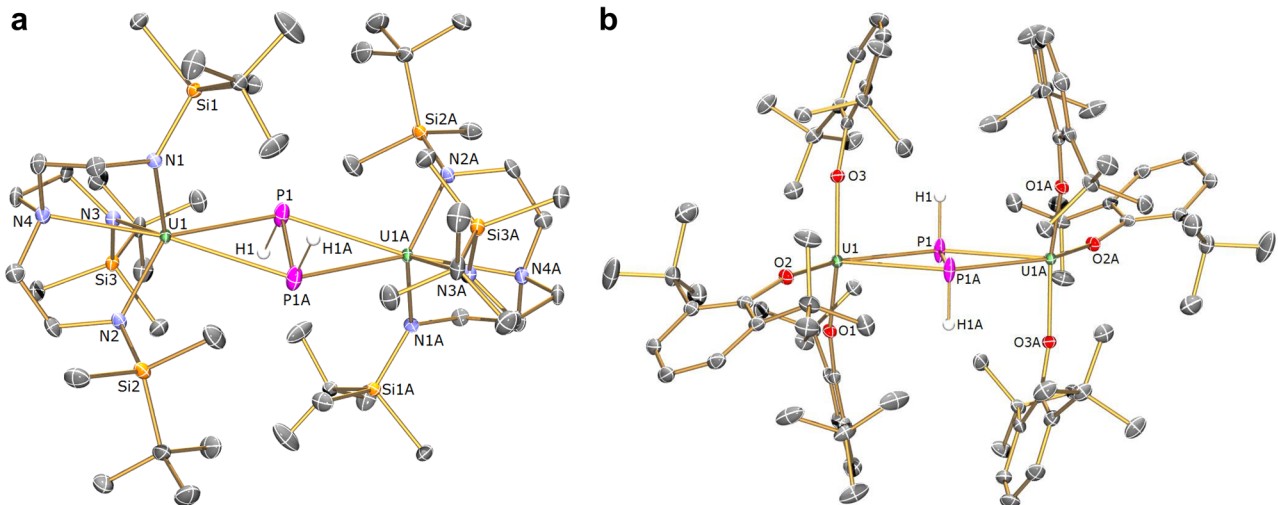

**Fig. 3 | Molecular structures of 8 and 9 at 100 K. a** structure of **8** with selected atom labels and displacement ellipsoids set at 40%. Pentane molecule in the lattice and hydrogen atoms except for H1 and H1A are omitted for clarity. **b** structure of **9** with selected atom labels and displacement ellipsoids set at 40%. Hydrogen atoms except for H1 and H1A are omitted for clarity. The structure of **6** is very similar in appearance except for the absence of H1 and H1A and shorter U-P and P-P distances.

intermediates. Whilst this is consistent with the experimentally observed P-P coupled products, this suggests the unfavourability of U adopting high (V/VI) oxidation states to form multiple polar covalent bonds to P. This underscores that high oxidation state U-phosphinidenes and -phosphides are intrinsically challenging targets to isolate.

## Results

### Synthesis

Treatment of **1** with **3** resulted in a red-brown solution, and work-up of the reaction mixture afforded brown crystals of the uranium(IV)-amide complex [(Tren$^{DMBS}$)U$^{IV}$(NMe$_2$)] (**4**) in crystalline yields of 32% (by U content), Fig. 2, demonstrating the fate of the NMe$_2$ group; however, no P$_2$ uranium product could be isolated from this reaction. This might be due to the less sterically protecting Tren$^{DMBS}$ ligand compared to Tren$^{TIPS}$, but the formation of **4** suggests that the reaction is analogous to the corresponding Tren$^{TIPS}$ reaction[37] and hence that a diphosphorus complex may be formed but is not isolated. In contrast, when **1** was reacted with **5**, Fig. 2, the new diphosphorus complex [(Ar*O)$_3$U$^{IV}$}$_2$(μ-η$^2$:η$^2$-P$_2$)] (**6**) was isolated as dark red crystals, but it was mixed with an unidentified brown solid which is presumably, analogously to the Tren$^{TIPS}$ reaction[37], the U$^{IV}$-amide [(Ar*O)$_3$U$^{IV}$(NMe$_2$)] (the analogue in this reaction of **IV** and **4**) or redistribution products thereof. Unlike the distinct solubilities of the diphosphorus and dimethylamido products supported by the Tren$^{TIPS}$ ligand, the solubility of **6** and the brown solid are essentially identical to each other, and so it was not possible to separate these two products, which hampered further characterization of **6**.

Addition of a solution of **3** in toluene to a solution of **2** in toluene led to a colour change from purple-red to bright red. Work-up of the reaction yielded two different sets of dark red crystals, with the major product being the diphosphene uranium(IV) complex [{(Tren$^{DMBS}$)U$^{IV}$}$_2${(μ-η$^2$:η$^2$-(HPPH)}] (**8**) in crystalline yield of 36% (by U content), and a small crop of diuranium(IV)-phosphinidiide complex [{(Tren$^{DMBS}$)U$^{IV}$}$_2$(μ-PH)] (**7**), Fig. 2. The crude $^1$H NMR (Supplementary Fig. 20) indicated a 2:1 ratio of **8** and **7** in this reaction, suggesting that P-P coupling of any intermediate U-PH moieties is more favourable than formal comproportionation reactions with **3**. When **5** was reacted with **2**, only the P-P coupled diphosphene complex [(Ar*O)$_3$U$^{IV}$}$_2${(μ-η$^2$:η$^2$-(HPPH)}] (**9**) was isolated as a red solid in a yield of 47%, Fig. 2. Inspection of crude reaction mixtures by $^1$H and $^{31}$P NMR

spectroscopies revealed only **9**, with no other U-containing complexes observed. This suggests that the aryloxide ligand is better, compared to the Tren$^{DMBS}$ ligand, at supporting the formation of P-P coupling to give a diphosphene versus comproportionation to give a phosphinidiide.

### Solid-state structures

The formulations of **6** (Supplementary Fig. 2), **8** and **9** were confirmed by their solid-state structures, Fig. 3a, b. The common salient feature of the three complexes is that the P$_2$ or HPPH unit is symmetrically side-on bound to two uranium centres. For the HPPH complexes, the H-atom positions were assigned as the *E*-isomers from the crystallographic Fourier difference maps, in agreement with the spectroscopic characterization data (see below). The U-P and P-P distances of 2.8910(10)/2.8271(11) and 2.048(2) Å in **6** are, respectively, comparable to those in [{(Tren$^{TIPS}$)U$^{IV}$}$_2$(μ-η$^2$:η$^2$-P$_2$)] (2.9441(12)/2.9446(12) and 2.036(2) Å)[37], confirming the presence of a U$_2$P$_2$ unit. The U-P distances in **8** (2.9535(17) and 3.0347(18) Å) and **9** (2.8791(8) and 2.9359(8) Å) are longer than the sum of the single bond covalent radii of U and P (2.81 Å)[49], but close to the side-on bound U-P distances in the above P$_2$ complexes as well as the terminal U-P distance of 2.883(2) and 2.8725(13) Å in [(Tren$^{TIPS}$)U$^{IV}$(PH$_2$)][50] and [(Tren$^{TCHS}$)U$^{IV}$(PH$_2$)] (Tren$^{TCHS}$ = {N(CH$_2$CH$_2$NSiCy$_3$)$_3$}$^{3-}$)[51], respectively. However, the marked difference is that the respective P-P distances of 2.211(4) and 2.1966(17) Å in **8** and **9** are longer than in **6**, being instead close to those in the reported d- and p-block diphospene complexes[18–21], and further compare well to the single bond covalent radii of P (2.22 Å) but are longer than twice the double bond covalent radii of P (2.04 Å)[49]. This suggests the presence of the diphosphane-1,2-diide form (HPPH)$^{2-}$ rather than the neutral diphosphene (HPPH)$^0$, consistent with prior predictions of the excellent π-acceptor character of HEEH moieties[52].

### Spectroscopic characterization

Unlike the related P$_2$ complex [{(Tren$^{TIPS}$)U$^{IV}$}$_2$(μ-η$^2$:η$^2$-P$_2$)][37] that is poorly soluble in common solvents, **8** and **9** have good solubility in benzene or toluene, facilitating the acquisition of solution-state spectroscopic characterization data. The $^1$H NMR spectra of **8** and **9** (Supplementary Figs. 25, 25, 29, and 30) in C$_6$D$_6$ exhibit 6 and 5 broad resonances over, for U$^{IV}$, quite large ranges of −150 to 6 ppm and −180 to 25 ppm, respectively. However, the majority of the $^1$H resonances for **8** and **9** fall in ranges spanning ~40 ppm which is normal for U$^{IV}$, and the

overall large chemical shift ranges are due to the HPPH proton resonances which resonate at −147.6 and −175.8 ppm, respectively. The $^{31}$P NMR spectra of **8** and **9** (Supplementary Figs. 28 and 31) show single broad resonances at 1065.4 and 844.4 ppm, respectively, which are rather different to those reported for the related complexes [(η$^5$-C$_5$H$_5$)$_2$Mo{μ-η$^2$:η$^2$-(HPPH)}] (203 ppm)[18,19], [(η$^5$-C$_5$H$_5$)$_2$Ta(H){μ-η$^2$:η$^2$-(HPPH)}] (−268.0 and −271.7 ppm)[20], and [{LGe}$_2${μ-η:η-(HPPH)}] (−182.5 ppm)[21], but within the range of related U$^{IV}$-PH complexes such as [{(Tren$^{TIPS}$)U$^{IV}$(μ-PH)(K-2.2.2-cryptand)}] (2460 ppm)[50] and [(Tren$^{TCHS}$)U$^{IV}$(PH)][(K-2.2.2-cryptand)] (2629 ppm)[51], which possess some U-P multiple bonding, and [(Tren$^{TIPS}$)U$^{IV}$(PH$_2$)] (595 ppm)[50] [(Tren$^{TCHS}$)U$^{IV}$(PH$_2$)] (605.9 ppm)[51], which have U$^{IV}$-P single bond interactions. The $^{29}$Si{$^1$H} NMR spectrum of **8** (Supplementary Fig. 27) exhibits a singlet at −42.4 ppm that is in the region of reported U$^{IV}$-$^{29}$Si chemical shifts[53].

The Raman spectrum of **8** (Supplementary Figs. 48, 49) exhibits a sharp peak at 445 cm$^{-1}$ that corresponds to the formal A$_1$ stretch of HPPH that is in good agreement with the computed P-P stretch of 427 cm$^{-1}$ and, as expected, lower than the P = P stretch of [{(Tren$^{TIPS}$) U$^{IV}$}$_2$(μ-η$^2$:η$^2$-P$_2$)] (589 cm$^{-1}$)[37]. By contrast, the Raman data of **9** (Supplementary Figs. 50, 51) shows very broad signals between 300 to 600 cm$^{-1}$, because of numerous ligand scattering modes, but the computed P-P stretching frequency lies inside that range (422 cm$^{-1}$) and is similar to **8**. The ATR-IR spectra of **8** and **9** (Supplementary Figs. 46, 47) display clear P-H stretching features at 2248 and 2265 cm$^{-1}$, which are close to the reported value for the free *trans*-structure of HPPH (2288 cm$^{-1}$)[8] and computed values of 2201 and 2253 cm$^{-1}$, respectively. From group theory considerations, the *Z*-isomer of HEEH would exhibit A and E stretching modes, but the *E*-isomer will only exhibit the E stretching mode since, possessing a centre of inversion, the A mode will be IR inactive; therefore, the presence of the *Z*-isomer can be ruled out in-line with the *E*-arrangement of HPPH determined in the solid-state structures of **8** and **9** and also consistent with the fact that [{(Tren$^{TIPS}$)U$^{IV}$}$_2${μ-η$^2$:η$^2$-(HAsAsH)}] also exclusively adopts the *E*-isomer[24].

The UV/Vis/NIR spectra of **8** and **9** (Supplementary Figs. 53, 54) exhibit broad absorptions in the range 5000–12,500 cm$^{-1}$, which are assigned as f-f transitions due to their intensities (< 100 M$^{-1}$cm$^{-1}$) and general patterns of their NIR regions that are characteristic of intraconfigurational absorptions of U$^{IV}$ ions[54]. Above 12,500 cm$^{-1}$ the spectrum is dominated by charge transfer bands.

## Magnetometric characterization

Variable-temperature SQUID magnetometry measurements were performed on powdered samples of **8** and **9** to confirm the U$^{IV}$ oxidation states required by their respective formulations (Supplementary Figs. 55, 57). The effective magnetic moments of **8** and **9** at 300 K are 4.37 and 3.79 μ$_B$ (3.09 and 2.68 μ$_B$ per U-ion) respectively, which decrease reaching 0.65 and 1.15 μ$_B$ (0.46 and 0.81 μ$_B$ per U-ion) respectively at 1.8 K and tending to zero, Fig. 4. Presenting a smooth decline of effective magnetic moment, the data for **8** are typical of U$^{IV}$[55], though the data for **9** essentially maintain the effective magnetic moment over most of the temperature range before a rapid decrease at low temperature, indicating a variance of $m_j$ states and their depopulation when replacing Tren$^{DMBS}$ with three aryloxides. We note that the χ$_M$ vs T plot for **8** exhibits a shoulder at low temperature that might be interpreted as resulting from anti-ferromagnetic coupling, but this is most likely due to single-ion crystal field effects[56]. Thus, the data for **8** indicate that it is a magnetic singlet at low-temperature, whereas for **9** the implication is that it is intermediate to magnetic singlet and triplet character at low temperature[57]. For both complexes, magnetization vs field data at 2 and 4 K present essentially linear responses over the 0-7 T range which do not saturate, confirming their U$^{IV}$ formulations (Supplementary Figs. 56, 58), and the magnetization value for **9** (1.1 N$_A$μ$_B$, 0.55 N$_A$μ$_B$ per U-ion) is approximately three times

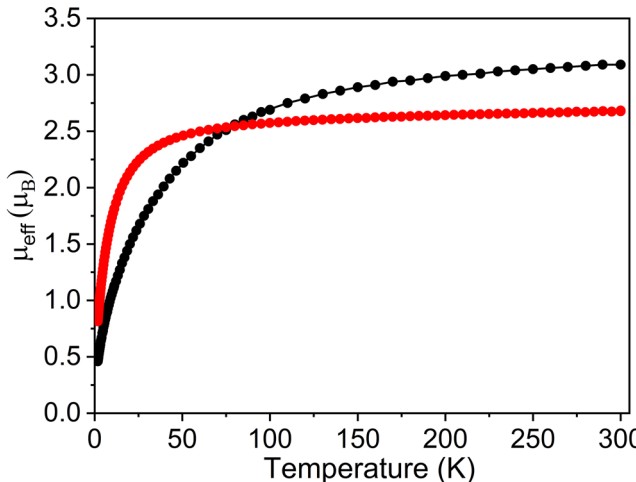

**Fig. 4 | Variable-temperature SQUID magnetometry data for 8 and 9.** Plots for **8** (black) and **9** (red) are of the effective magnetic moment μ$_{eff}$ (μ$_B$) per ion over the temperature range 1.8–300 K. Lines are a guide to the eye only.

that of **8** (0.35 N$_A$μ$_B$, 0.175 N$_A$μ$_B$ per U-ion) at 7 T, consistent with the low temperature effective magnetic moment data.

## Quantum chemical bonding analysis

In order to probe the bonding of the U-P and P-P linkages in **6-9** we performed quantum chemical density functional theory calculations on the whole structures (Supplementary Figs. 59–62, Supplementary Tables 1 and 2) using triple-ζ-plus polarization all-electron basis sets and the general gradient approximation BP86 functional. The geometry optimized structures of the quintet spin formulations are in good agreement with the solid-state structures, consistent with the quintet spin formulation of [{(Tren$^{TIPS}$)U$^{IV}$}$_2$(μ-η$^2$:η$^2$-P$_2$)] being lower in energy than the septet spin state[37], and also consistent with the magnetic characterization data. Hence, we conclude that the calculations provide a qualitative description of the electronic structures of **6-9**.

For **8** and **9**, computed Multipole Derived Charge (MDC) analysis reveals U/P MDC$_q$ charges of 2.89/−1.14 and 2.52/−0.99, respectively. The HPPH moieties of **8** and **9** exhibit charges of −2.71 and −2.20 overall, respectively, and noting that the N$_{amido}$ charges average −1.52 the MDC$_q$ data are consistent with the HPPH units being assigned diphosphane-1,2-diide, rather than diphosphene, character. The data are similar to those of [{(Tren$^{TIPS}$)U$^{IV}$}$_2${μ-η$^2$:η$^2$-(HAsAsH)}] (MDC$_q$ U/As = 3.20/−1.12)[24], but overall are less polarized for **8** and certainly **9**, suggesting that HPPH is a more effective donor to U than HAsAsH, or put another way HAsAsH is a better acceptor than HPPH, as predicted theoretically[52]. We also note that the U/P charges are lower in **9** than **8**, suggesting that the 3 × Ar*O ligand set leaves U more electron deficient than with Tren$^{DMBS}$, and hence the U-P interactions are consequently more fully developed in **9** compared to **8**. The U-P/P-P Nalewajski-Mrozek bond orders for **8** and **9** of 0.68/1.10 and 0.69-0.82/1.01 are also consistent with the bonding picture suggested by the MDC$_q$ charges, and again support the view that HPPH is present as its dianion form with a P-P single bond due to population of the P-P π- and π*-orbitals.

The top four most energetic electrons in **8** are quasi-degenerate (0.05 eV range) singly occupied α-spin HOMO to HOMO−3, which are essentially pure 5f character (Supplementary Fig. 61). The α-spin HOMO−4 (and β-spin equivalent) represent the frontier U-P interactions, and are, formally, U 6d to HPPH 3p π* back-bonding that is 60:40 P:U character, Fig. 5. A similar picture is presented in the frontier bonding manifold of **9**. However, the relevant molecular orbitals (HOMO−4 and −5, Supplelentary Fig. 62, and β-spin manifold

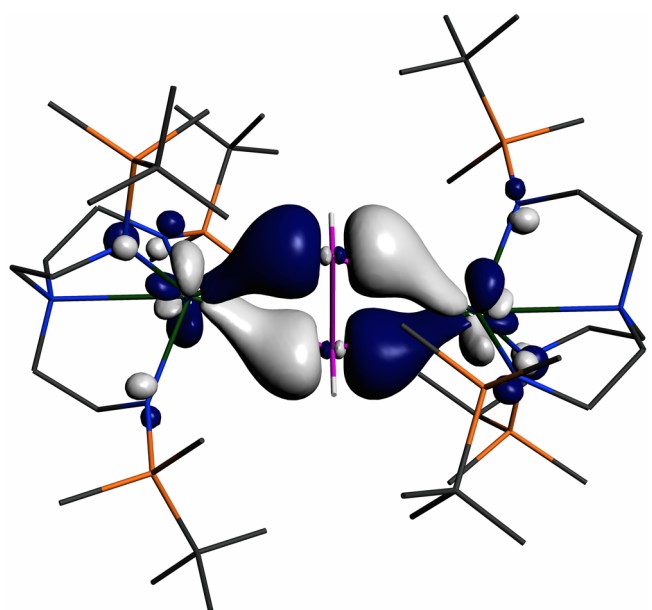

**Fig. 5 | The α-spin HOMO − 4 (377a, −4.242 eV) representation of the U-P interaction in 8.** Computed at the BP86 LDA VWN ZORA TZP all-electron level of theory. The β-spin equivalent is very similar. Non-P-H H-atoms are omitted for clarity. Key: U green, P magenta, Si orange, N blue, C grey, H white. This bonding combination is also found for **9** but there it is mixed in with aryl orbital coefficients.

equivalents) are mixed with Ar*O orbital coefficients, and so whilst the gross bonding is clear a detailed discussion is precluded.

For **6**, the U/P MDC$_q$ charges are 2.83/−1.17 and the P-P Nalewajski-Mrozek bond order is 1.4, which can be compared to corresponding values for [{(Tren$^{TIPS}$)U$^{IV}$}$_2$(μ-η$^2$:η$^2$-P$_2$)][37] of 3.55/−1.4 and 1.50 and 3.40/−1.51 and 1.36 for [{(Tren$^{TIPS}$)U$^{IV}$}$_2$(μ-η$^2$:η$^2$-P$_2$)]$^-$ which contains P$_2^{3-\bullet}$[42]. This supports the assignment of the diphosphorus unit in **6** as P$_2^{2-}$ and reinforces that the 3 × Ar*O ligand set leaves U more electron deficient than with Tren$^{DMBS}$, and hence the U-P interactions are resultantly more fully developed in **6** compared to **II**. We note that the molecular orbitals of **6** (Supplementary Fig. 59) suggest a very similar bonding situation to **II**, that is three α-spin f-electrons, then an α-spin electron that is a mixture of 5f and P$_2$ π* character (the B(π$_{g\perp}$) δ-symmetry molecular orbital), and then the A(π$_{g=}$) in-plane U-P π-bond; however, variable mixing of Ar*O orbital coefficients into the latter two types of molecular orbitals prevents detailed discussion. Lastly, for **7** the bonding picture suggests a relatively polar bonding picture (MDC$_q$ U/P = 2.88 (av.)/−2.54), with a 3-centre-2-electron U-P-U bond supplemented by dative 3-centre-2-electron P π-donation to both U ions (Supplementary Fig. 60).

## Quantum chemical reaction profile analyses

In order to gain insights into the formation of **4, 6, 7, 8** and **9**, DFT intrinsic reaction coordinate profile calculations (B3PW91), including dispersion corrections, were examined (Supplementary Tables 3-6). We firstly examine the formations of **8**, Fig. 6, and **9** (Supplementary Fig. 63) since they exhibit the same *trans*-HPPH unit. For **8**, the reaction begins with the coordination of **2** to trivalent **3**, leading to the thermodynamically stable intermediate **Int1** (−5.4 kcal mol$^{-1}$). It is interesting to note that like CO$_2$ coordination to U$^{III}$[58], the formation of **Int1** involves a single electron transfer from the U-centre (formal oxidation to U$^{IV}$ by the 7λ$^3$-phosphadibenzonorbornadiene), resulting in formation of a polar covalent U-P bond with concomitant cleavage one of the P-C bonds and localization of a radical on the anthracene ring. This is evidenced by the unpaired spin density values of 2.2 at the U-ion (in accordance with a 5f$^2$ U$^{IV}$, the corresponding value for 5f$^3$ U$^{III}$ **3** is 3.1)

and 0.62 at the anthracene (in line with a radical formation). **Int1** readily extrudes anthracene via a low-lying transition state (**TS1**) with an associated enthalpy barrier of only 12.9 kcal mol$^{-1}$. As highlighted by the unpaired spin values of **TS1**, Fig. 6, the P-C bond breaks homolytically and also the delocalized π-system of anthracene begins to form. This produces **Int2** (−9.2 kcal mol$^{-1}$), which is a Van der Waals adduct of anthracene to a uranium phosphinido complex. The unpaired spin analysis on the latter clearly indicates S = 3/2 open-shell radical character, with the U and P atoms exhibiting unpaired spins of 2.22 and 0.92, respectively, rather than radical combination to formally oxidize the U-ion to give a S = 1/2 U$^V$ = PH linkage, which is calculated to be 7.5 kcal mol$^{-1}$ less stable than the S = 3/2 form. The radical character of **Int2** accounts for the formation of **7** and **8** via radical coupling of **Int2** with either itself or **Int1** (formation of **8**) or with **3** (formation of **7**), which are thermodynamically very favourable processes, Fig. 6. The formation of **9** from the reaction of **2** and **5** follows a similar pathway (Supplementary Fig. 63), with the main differences being that the barrier is higher (14.0 kcal mol$^{-1}$), the uranium phosphinido radical $^{ArO}$**Int2** is even more unstable than **Int2** (by 15 kcal mol$^{-1}$), and the S = 1/2 form of $^{ArO}$**Int2** is more unstable than the S = 3/2 form by 12.1 kcal mol$^{-1}$. Also, the difference in barrier height is associated with the steric hindrance around **5**, which renders the coordination and subsequent reaction steps slightly more challenging. These factors combine to enforce $^{ArO}$**Int2** reacting with itself to give **9** rather than being long-lived enough to also react with **5** to make the aryloxide phosphinidiide analogue of **7**.

Since the reactivity of **2** with **3** or **5** follow similar pathways, it could be surmised that **1** reacts with **3** or **5** in similar reactions since the respective isolations of **4** and **6** are both analogous to the products **II** and **IV** that were both isolated from the corresponding reaction of **I** with **1**. The formation of **4**, Figs. 7, and 6 (Supplementary Fig. 64) have thus been investigated computationally using the same methodology. For the reaction of **3** with **1**, like that of **3** with **2**, reductive coordination of **1** to **3** (**Int3**) produces a polar covalent U$^{IV}$-P bond and P-C cleavage to produce an anthracene-based radical, which is followed by P-C bond breaking (**TS2**) and the formation of the uranium phosphinido radical (**Int4**). The associated barrier is only 10.7 kcal mol$^{-1}$ and the formation of **Int4** is exothermic by 4.1 kcal mol$^{-1}$ overall at that point of the reaction coordinate. Next, **Int4** reacts with **3** to form **Int5**, which is analogous to **7**, and related **III**, and the formation of **Int5** is favourable (−19.8 kcal mol$^{-1}$). In **Int5**, the phosphinidiide ligand is (κ$^2$-P,η$^2$-PN) bonded. Therefore, the P-N bond can be broken via **TS3** with an accessible activation barrier of 21.7 kcal mol$^{-1}$, and this P-N bond breaking step appears to be the rate determining step of the reaction. Following the intrinsic reaction coordinate, **4** is eliminated producing **Int6**, which has U and P spin densities of 2.27 and 1.81, respectively, and thus this is a S = 2 open-shell quintet U$^{IV}$-phosphide complex with non-interacting U and P triplet diradicals. Complex **Int6** would readily dimerize due to its radical nature to yield the more stable (by 87.4 kcal mol$^{-1}$) complex **6'**, which is the equivalent of **6** and **II**. Regarding the quintet form of **Int6**, whilst radical combination could in principle produce the S = 1 (open-shell triplet U$^V$ = P$^\bullet$) and S = 0 (closed-shell singlet U$^{VI}$≡P) forms, they are computed to be 4.7 and 9.3 kcal mol$^{-1}$ less stable than the S = 2 form, respectively; attempts to compute the open- shell singlet form of **Int6** proved to be intractable, suggesting significant instability of that species. The formation of **6** (and **4'**) is found to follow a similar pathway (Supplementary Fig. 64), and as already found for the formation of **9** compared to **8**, the barriers are higher, but still accessible, for the formation of **6** and **4'** than for the formation of **4** and **6'**. Indeed, the two transition state barriers for **6** and **4'** are 18.9 and 26.8 kcal mol$^{-1}$, respectively, that are ~8–10 kcal mol$^{-1}$ higher than for the formation of **4** and **6'**. This increase is similar to that found for the reaction of **3** or **5** with **2** and is associated with the greater steric hindrance associated with **5** compared to **3**. Lastly, the intermediate $^{ArO}$**Int6** is found to be most stable in its S = 2

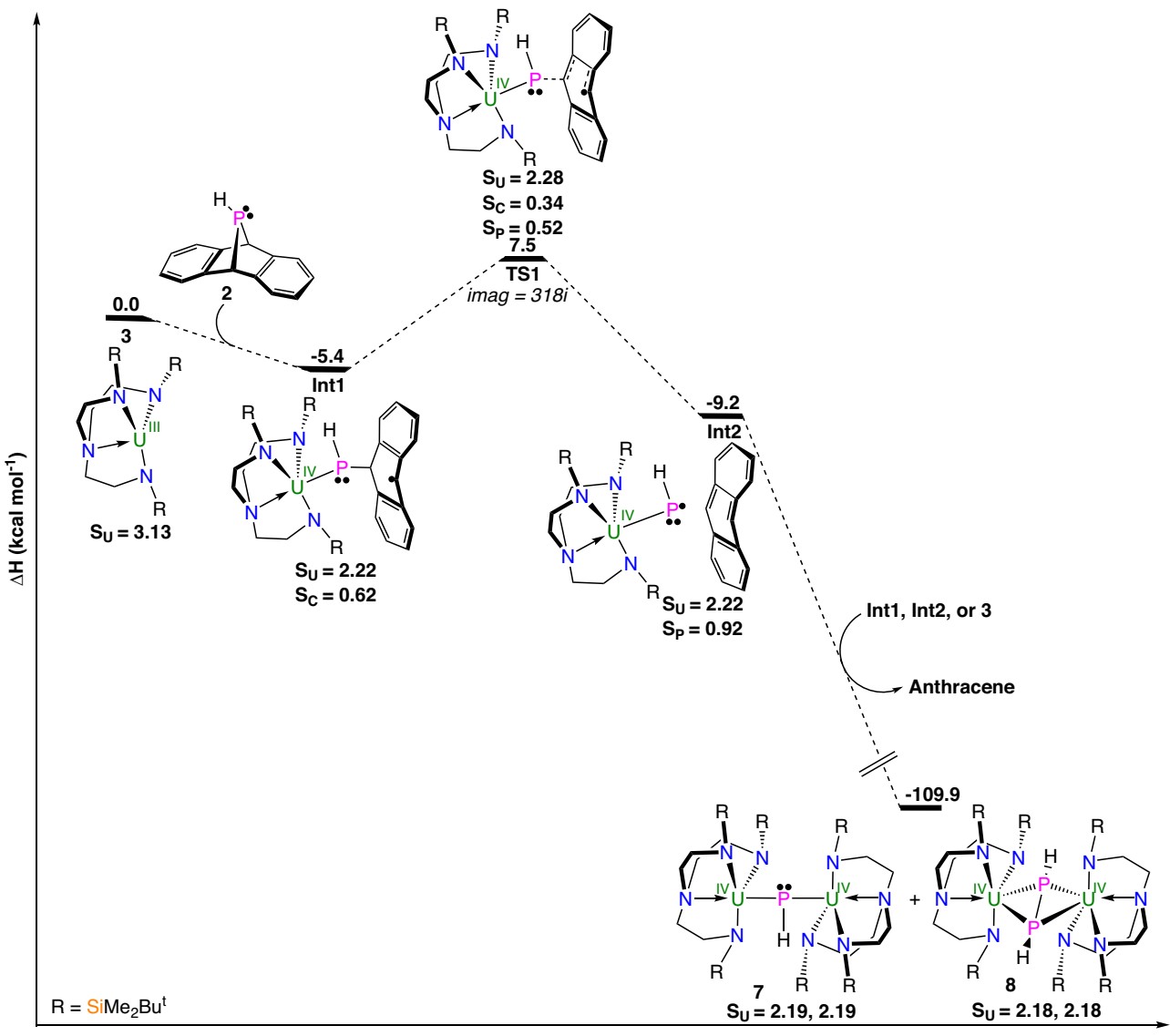

**Fig. 6 | Computed enthalpy profile at room temperature for the formation of 7 and 8 from the reaction of 3 and 2.** Computed with B3PW91, small core pseudopotential Stuttgart basis set for U and P atoms with additional polarization functions for P atoms, Pople 6-31 G** basis sets for other atoms, and Grimme's D3 dispersion with Becke-Johnson damping. The enthalpies are given in kcal mol⁻¹. The unpaired spin density values are reported for each intermediate. The analogous reaction of **5** with **2** to give **9**, Supplementary Fig. 63, is similar.

form, like **Int6**, with the $S = 1$ and $S = 0$ spin forms being less stable by 4.7 and 13.7 kcal mol⁻¹, respectively.

## Discussion
By examining the reactivity of 7λ³-R-phosphadibenzonorbornadiene with sterically varied uranium(III) complexes, we have introduced the diphosphene linkage to f-element chemistry and also prepared new diuranium-diphosphorus and -phosphinidiide complexes. The isolation of **8** and **9** demonstrates that **2** is an excellent PH-group transfer reagent and via P-P coupling an approach to develop a novel synthetic method for constructing formal parent diphosphene complexes.

Quantum chemical calculations show that the coordinated HPPH units are present in their phosphane-1,2-diide forms. This finding is consistent with prior predictions of the strong π-acceptor nature of HPPH[52]. The calculations also highlight the important role of U 5f-orbital participation in π-bonding in these complexes, which are

reminiscent and isoelectronic analogues of f-element π-alkene complexes, which themselves remain rare[59–62].

The experimental and computational observations of this study permit us to comment on factors that are key to determining the outcomes of each reaction, and also to confirm that **1** and **2** effect the formal transfer of phosphinidene groups. For the reactions of **1** with **I**, **3**, and **5** the outcome appears to be the same in all cases, where diphosphorus and amide products form. Clearly, none of Tren^DMBS, Tren^TIPS, or 3 × Ar*O are too big nor too small to prevent side-on binding of P₂, and the P-N bond is robust enough to support, and indeed as a π-donor promote, the initial transfer of the PR group but is then labile enough to be cleaved away (forming a favourable U-N bond) to give a radical phosphide that dimerizes rather than stopping at a bridging phosphinidiide stage, which from prior work would be sterically accessible. That the latter step occurs rather than making isolable U^VI≡P derivatives is a direct consequence of the preference for a radical intermediate that favours P-P coupling, which can be

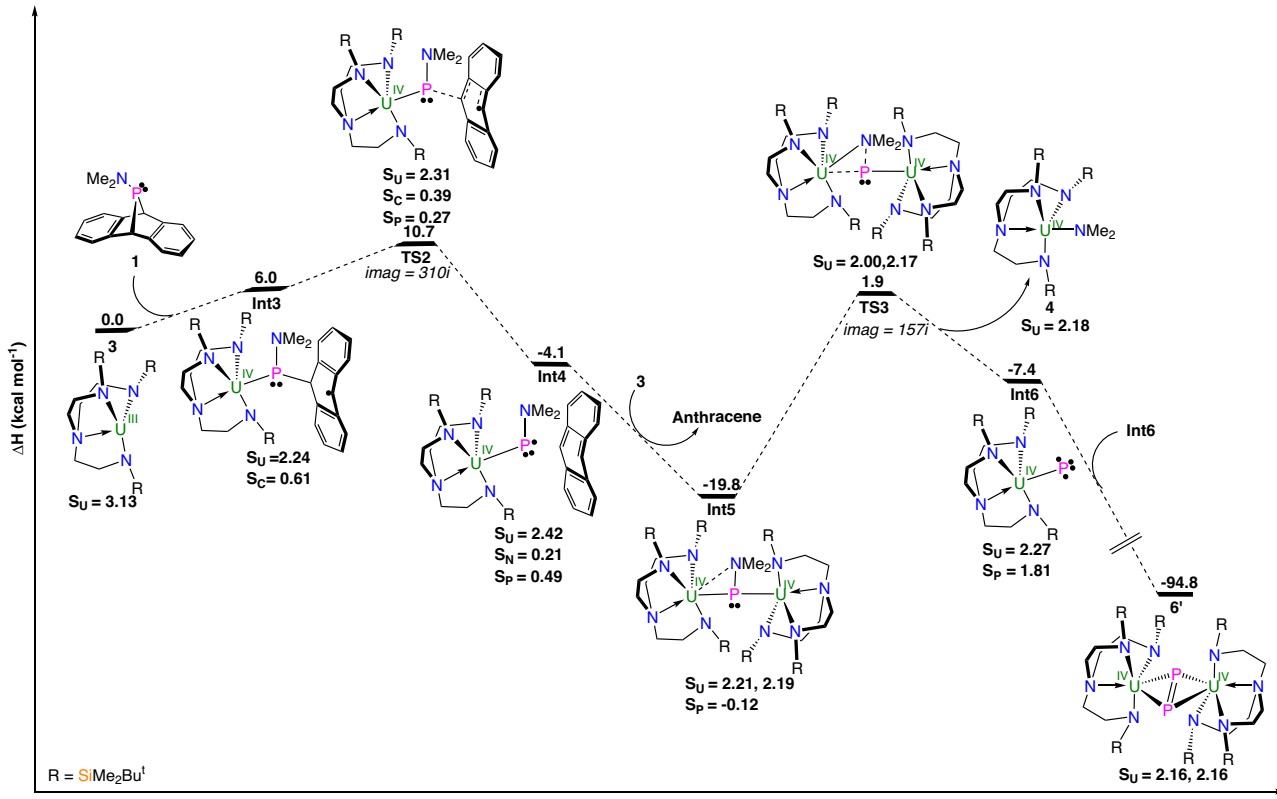

**Fig. 7 | Computed enthalpy profile at room temperature for the formation of 4 and 6′ from the reaction of 3 and 1.** Computed with B3PW91, small core pseudopotential Stuttgart basis set for U and P atoms with additional polarization functions for P atoms, Pople 6-31 G** basis sets for other atoms, and Grimme's D3 dispersion with Becke-Johnson damping. The enthalpies are given in kcal mol⁻¹. The unpaired spin density values are reported for each intermediate. The analogous reaction of **5** with **1** to give **4′** and **6**, Supplementary Fig. 64, is similar.

contrasted to the fact terminal M^VI≡P (M = Mo, W) and bridging U-/Th phosphides are isolable due to better spatial and energy matching of the frontier M and P orbitals and electrons[29,63–72]. Where reactions of **2** with **I**, **3**, and **5** are concerned, ancillary ligand sterics play a more decisive role. Whereas only **9** forms, presumably because 3 × Ar*O ligands at U leaves enough space to allow dimerization, for the slightly more sterically demanding Tren^DMBS both **7** and **8** form suggesting a fine balance between phosphinidiide and diphosphene linkage formation, but for the sterically even more hindered Tren^TIPS only the bridging phosphinidiide can form. Given the commonality of intermediate phosphinidene radicals that convert to phosphinidiides, it is telling that the P-H species do not undergo P-H cleavage the way P-N cleavage occurs, and presumably this is because the formation of U-H is not sufficiently favourable enough compared to U-NMe₂.

In previous work, we found that one-electron oxidation of [U^IV(Tren^TIPS)(NH)]⁻ to [U^V(Tren^TIPS)(NH)] resulted in disproportionation to [U^IV(Tren^TIPS)(NH₂)] and [U^VI(Tren^TIPS)(N)][73,74]. By contrast, the analogous one-electron oxidation of [U^IV(Tren^TIPS)(PH)]⁻ results in the isolation of [U^IV(Tren^TIPS)(PH₂)] and **II**[51], but the latter can be viewed as dimerized [U^VI(Tren^TIPS)(P)] and hence that reaction also has the initial appearance of a uranium disproportionation reaction. However, a striking feature of the computed reaction profiles in this study is the dominance of the U^IV oxidation state and the appearance of radical species rather than the all-electron paired high oxidation state U analogues. This is certainly consistent with HSAB theory where the soft P evidently does not stabilize high oxidation states of U. Thus, this work sheds light on three key aspects: (i) the redox chemistry is driven by the P and not U centres; (ii) the presence of open-shell radical species in these reactions, rather than closed-shell singlets, suggests that

attempts to isolate high oxidation state phosphinidene and phosphide complexes of U^V and U^VI is inherently challenging[75], which can again be traced back to HSAB drivers; (iii) the prevalence of radical intermediates is responsible for the formation of P-P catenated derivatives. Although high oxidation state U will have somewhat contracted valence orbitals, the P-anion valence orbitals are reasonably diffuse, and so we suggest that the prevalence of open-shell radical over closed-shell singlet species principally originates from poor energetic matching of U and P frontier orbitals rather than excessively poor spatial overlap. Overall, it is clear that there are common reaction steps in all the reactions studied here, but the precise diverged outcomes are controlled by the ancillary ligands, the radical nature of intermediates, and the phosphinidene substituent.

## Methods

### General experimental details

All manipulations were carried out under an inert atmosphere of dry N₂ using Schlenk techniques or an MBraun UniLab glovebox. Solvents were dried by passage through activated alumina towers, or for benzene distilled from K, and degassed before use. Solvents were stored over K-mirrors except for ethers which were stored over activated 4 Å sieves. D-solvents were distilled from K, degassed by three freeze-pump-thaw cycles, and stored under N₂ prior to use. ClSiMe₂Bu^t was distilled from Mg, degassed by three freeze-pump-thaw cycles and stored under N₂. Magnesium powder was activated prior to use as described below. Sodium bis(trimethylsilyl)amide was recrystallised from a saturated pentane solution prior to use. Triphenylborane and anthracene were dried under dynamic vacuum (1 × 10⁻³ mbar) for 24 h prior to use. Elemental potassium was freed from oxides and washed

with hexane to remove mineral oil prior to use. Other chemicals were purchased from commercial sources and used as received. Depleted $UO_3$ was supplied by the National Nuclear Laboratory. The compounds [(Anthracene)Mg(THF)$_3$], PDBN-NMe$_2$ (**1**) (PDBN = 7$\lambda^3$-phosphadibenzonorbornadiene), PDBN-H (**2**), PDBN-(BPh$_3$)Na(OEt$_2$)$_2$, PDBN-Bu$^t$[38,39,76,77], KC$_8$[36,78], UI$_3$(1,4-dioxane)$_{1.5}$[79], [U{N(SiMe$_3$)$_2$}$_3$][79], U$^{IV}$Cl$_4$[36,80], Tren$^{DMBS}$Li$_3$(THF)$_3$[81], [(Tren$^{DMBS}$)U$^{IV}$(Cl)][82], [(Tren$^{DMBS}$)U$^{IV}$(I)][82], [{K(Toluene)][(Tren$^{DMBS}$)U$^{Cyclomet}$]}$_2$][83], [(Tren$^{DMBS}$)U$^{III}$] (**3**)[84], [(Ar*O)$_3$U$^{III}$] (**5**)[48] were prepared by the modified procedures described below.

Single crystals were examined variously using either a Rigaku XtaLab Synergy or Rigaku FR-X diffractometer, each equipped with a HyPix 6000HE photon counting pixel array detector with mirror-monochromated Cu Kα (λ = 1.5418 Å) radiation. Intensities were integrated from a sphere of data recorded on narrow (0.5° (Synergy) or 1.0° (FR-X)) frames by ω rotation. Cell parameters were refined from the observed positions of all strong reflections in each data set. Gaussian grid face-indexed absorption corrections with a beam profile correction were applied. The structures were solved by dual methods using SHELXT[85] and all non-hydrogen atoms were refined by full-matrix least-squares on all unique $F^2$ values with anisotropic displacement parameters with exceptions noted in the respective cif files. Except where stated for P-H hydrogens, all hydrogen atoms were refined with constrained geometries and riding thermal parameters; $U_{iso}$(H) was set at 1.2 (1.5 for methyl groups) times $U_{eq}$ of the parent atom. The largest features in final difference syntheses were close to heavy atoms and were of no chemical significance. CrysAlisPro was used for control and integration[86], and SHELXL and Olex2 were employed for structure refinement[87,88]. ORTEP-3 and POV-Ray were employed for molecular graphics[89,90].

$^1$H, $^{29}$Si{$^1$H}, and $^{31}$P spectra were recorded on a Bruker 400 MHz spectrometer operating at 400, 79, and 162 MHz, respectively; chemical shifts are quoted in ppm and are relative to TMS ($^1$H, $^{29}$Si), and 85% H$_3$PO$_4$ ($^{31}$P), respectively. ATR-IR spectra were recorded on a Bruker Alpha spectrometer with a Platinum-ATR module in the glovebox. Raman spectra were recorded on a Horiba XploRA Plus Raman microscope with a 638 nm laser with a power of 1.5 mW. The power was adjusted using a power filter for each complex to inhibit sample decomposition. UV/Vis/NIR spectra were recorded on a Perkin Elmer Lambda 750 spectrometer. Data were collected in a 1 mm path-length cuvette and were run versus the appropriate solvent. Variable-temperature magnetic moment data were recorded in an applied direct current (DC) field of 0.5 Tesla on a Quantum Design MPMS3 superconducting quantum interference device magnetometer using recrystallized powdered samples. Measurements were performed in dc scan mode using 40 mm scan length and 6 s scan time. Samples were carefully checked for purity and data reproducibility between independently prepared batches. Samples were crushed with a mortar and pestle under an argon atmosphere and immobilized in an eicosane matrix within 400 MHz Wilmad borosilicate NMR tubes to prevent sample reorientation during measurements. The tube was flame-sealed under dynamic vacuum (1 × 10$^{-3}$ mbar) to a length of ~3 cm and mounted in the centre of a drinking straw, with the straw fixed to the end of an MPMS 3 sample rod. Care was taken to ensure complete thermalization of the sample before each data point was measured by employing delays at each temperature point as well as a slow cooling rate (5 K/min from 300 to 100 K; 2.5 K/min from 100 to 50 K; 1 K/min from 50 to 1.8 K). The sample was held at 2 K for 30 min before isothermal magnetization measurements to account for slow thermal equilibration of the sample. Diamagnetic corrections were applied using tabulated Pascal constants. Measurements were corrected for the effect of the blank sample holders (flame sealed Wilmad NMR tube and straw) and eicosane matrix. CHN microanalyses were carried out on a Flash 2000 elemental analyser.

## Modified procedure for the preparation of [(anthracene)Mg(THF)$_3$]

Prior to use, the magnesium powder needs to be activated. Under an argon atmosphere, Mg powder (2.40 g, 100 mmol) was heated to 250 °C under vacuum ( ~ 1 × 10$^{-3}$ mbar) for 4 hours. The reaction vessel was then allowed to cool to room temperature, before THF (300 mL) was added along with a few drops of 1,2-dibromoethane. The resultant suspension was stirred for 12 hours, before solid anthracene (21.36 g, 120.00 mmol) was added in a portion-wise manner using a solid-addition funnel. The reaction mixture was then stirred for four days, during which time there was precipitation of an orange solid which was separated by filtration. Volatiles were then removed *in vacuo* before the solid was washed with THF (4 × 50 mL) and dried *in vacuo* to afford [(Anthracene)Mg(THF)$_3$] as an orange solid, which was used without further purification. Yield: 36.12 g, 86%. The poor solubility of [(Anthracene)Mg(THF)$_3$] in hydrocarbon, arene, and ethereal solvents precluded the acquisition of NMR spectroscopic data. ATR-IR v/cm$^{-1}$: 3029 (m), 2948 (m), 2887 (m), 1568 (m), 1456 (s), 1434 (w), 1362 (s), 1245 (s), 1199 (w), 1173 (s), 1143 (w), 1103 (s), 1020 (s), 915 (m), 872 (s), 839 (w), 806 (s), 778 (m), 754 (s), 714 (s), 674 (w), 571 (m), 428 (s).

## Modified procedure for the preparation of PDBN-NMe$_2$ (1)

Me$_2$NPCl$_2$ (5.00 g, 34.26 mmol) was dissolved in THF (300 mL) and cooled to −78 °C. To this, [(Anthracene)Mg(THF)$_3$] (14.35 g, 34.26 mmol) was added in a portion-wise manner with vigorous stirring and a 20-minute delay between each portion added (approx. 8 portions in total). Over the course of the addition, there was a colour change to orange and then pale yellow. The reaction was stirred at −78 °C for four hours, before being allowed to warm to room temperature and volatiles removed *in vacuo* to afford a pale-yellow residue. Toluene (200 mL) was added the mixture and slurried for 10 min before being filtered through a Celite-padded coarse porosity frit to yield a yellow solution. Volatiles were then removed *in vacuo* before soluble residues were extracted through the addition of DCM (100 mL). The resultant suspension was stored at −30 °C for 6 h, resulting in the precipitation of unwanted side-products. The mixture was filtered, and volatiles were then removed *in vacuo* to obtain a yellow solid. Soluble residues were extracted with Et$_2$O (60 mL) and filtered to yield a yellow solution, which was stored at −30 °C for 24 hours to afford **1** as a pale yellow crystalline solid. Yield: 3.13 g, 36%. $^1$H NMR (400 MHz, C$_6$D$_6$, 298 K): δ (ppm) 2.23 (d, $^2J_{PH}$ = 7.5 Hz, 6H, NC$H_3$), 4.12 (d, $^2J_{PH}$ = 13.0 Hz, 2H, C$H$), 6.83 (m, 2H, Ar-$H$), 7.03 (m, 2H, Ar-$H$), 7.09 (m, 2H, Ar-$H$), 7.28 (m, 2H, Ar-$H$). ATR-IR v/cm$^{-1}$: 3059 (w), 3012 (w), 2973 (w), 2919 (w), 2886 (w), 2841 (w), 2795 (w), 1465 (w), 1448 (s), 1408 (w), 1262 (s), 1192 (w), 1179 (w), 1155 (m), 1104 (w), 1072 (s), 1056 (w), 1017 (w), 965 (s), 882 (s), 789 (s), 756 (s), 744 (s), 724 (s), 679 (m), 668 (m), 621 (w), 603 (s), 575 (w), 514 (s), 473 (w), 454 (w), 434 (w), 416 (w).

## Modified procedure for the preparation of PDBN-H (2)

In the strict absence of light, a solution of di-iso-butylaluminum hydride (1 M in hexane, 20.00 mL, 20.00 mmol) was added to a stirring solution of **1** (2.00 g, 8.00 mmol) in toluene (10 mL) at −78 °C. The mixture was allowed to warm to room temperature, during which time there was the precipitation of a solid resulting in the formation of a milky-white suspension. After two hours, hexane (50 mL) was added resulting in the rapid precipitation of more solid. The reaction mixture was carefully filtered to isolate the precipitate, and removal of volatiles *in vacuo* afforded **2** as a white solid. Yield: 1.48 g, 88%. $^1$H NMR (400 MHz, C$_6$D$_6$, 298 K): δ (ppm) 3.86 (d, $^2J_{PH}$ = 14.2 Hz, 2H, C$H$), 5.41 (d, $^1J_{PH}$ = 162.0 Hz, 1H, P$H$), 6.72 (m, 2H, Ar-$H$), 6.85 (m, 2H, Ar-$H$), 6.97 (m, 2H, Ar-$H$), 7.10 (m, 2H, Ar-$H$). ATR-IR v/cm$^{-1}$: 3050 (w), 3013 (w), 2999 (w), 2236 (P-H, s), 1463 (w), 1447 (s), 1182 (m), 1169 (m), 1151 (s), 1107 (w), 1093 (s), 1054 (s), 1013 (s), 998 (w), 981 (w), 956 (w), 937 (w),

906 (w), 881 (s), 766 (s), 736 (s), 725 (m), 699 (s), 629 (s), 595 (s), 492 (s), 474 (s), 432 (m).

## Modified procedure for the preparation of PDBN·(BPh$_3$)Na(OEt$_2$)$_2$

In the strict absence of light, a solution of sodium bis(trimethylsilyl)amide (0.92 g, 5.00 mmol) in diethyl ether (10 mL) was added to a stirring solution of **2** (1.26 g, 5.00 mmol) and triphenylborane (1.21 g, 5.00 mmol) in diethyl ether (30 mL) at −78 °C. The mixture was allowed to warm to room temperature, during which time there was the precipitation of a solid resulting in the formation of an off-white suspension. After one hour, the reaction mixture was carefully filtered to isolate the precipitate, and removal of volatiles *in vacuo* afforded an off-white solid which was washed with Et$_2$O (2 × 5 mL) to yield PDBN·(BPh$_3$)Na(OEt$_2$)$_2$ as a colourless solid. Yield: 2.43 g, 78%. $^1$H NMR (400 MHz, C$_6$D$_6$, 298 K): δ (ppm) 0.96 (t, 12H, OEt$_2$-C$H_3$), 3.10 (q, 8H, OEt$_2$-C$H_2$), 3.76 (d, $^2J_{PH}$ = 12.8 Hz, 2H, C$H$), 6.44−6.49 (m, 4H, Ar-$H$), 6.64 (m, 2H, Ar-$H$), 6.89−6.96 (m, 5H, Ar-$H$), 7.05 (m, 6H, Ar-$H$), 7.23 (d, 6H, Ar-$H$). ATR-IR v/cm$^{-1}$: 3058 (br, m), 2972 (br, m), 2930 (w), 2866 (br, w), 1581 (m), 1479 (m), 1464 (w), 1448 (s), 1427 (m), 1383 (m), 1351 (w), 1303 (w), 1264 (w), 1176 (w), 1152 (m), 1083 (s), 1031 (w), 927 (w), 843 (w), 782 (s), 763 (w), 702 (s), 649 (m), 629 (m), 601 (m), 573 (w), 507 (s).

## Modified procedure for the preparation of PDBN·Bu$^t$

$^t$BuPCl$_2$ (1.58 g, 10.00 mmol) was dissolved in THF (100 mL) and cooled to −78 °C. To this, [(Anthracene)Mg(THF)$_3$] (4.18 g, 10.00 mmol) was added in a portion-wise manner with vigorous stirring and a 20-minute delay between each portion added (~4 portions in total). Over the course of the addition, there was a colour change to orange and then yellow/green. The reaction was stirred at −78 °C for two hours, before being allowed to warm to room temperature and volatiles removed *in vacuo* to afford a pale-yellow residue. Toluene (100 mL) was added the mixture and slurried for 10 minutes before being filtered through a Celite-padded coarse porosity frit to yield a yellow solution. Volatiles were then removed *in vacuo* before soluble residues were extracted through the addition of DCM (50 mL). The resultant suspension was stored at −30 °C for 6 h, resulting in the precipitation of unwanted side-products. The mixture was filtered, and volatiles were then removed *in vacuo* to obtain a yellow solid. Soluble residues were extracted with hexane (60 mL) and filtered to yield a yellow solution, which was concentrated (approx. 15 mL) and stored at −30 °C for 24 h to afford PDBN·Bu$^t$ as a colourless crystalline solid. Yield: 0.43 g, 16%. $^1$H NMR (400 MHz, C$_6$D$_6$, 298 K): δ (ppm) 0.77 (d, NC$H_3$, 9H), 3.90 (d, 2H, C$H$), 6.74−6.76 (m, 2H, Ar-$H$), 6.92−6.96 (m, 4H, Ar-$H$), 7.18−7.20 (m, 2H, Ar-$H$). ATR-IR v/cm$^{-1}$: 3048 (w), 1620 (m), 1533 (w), 1448 (s), 1314 (m), 1271 (w), 1181 (w), 1146 (m), 1119 (w), 1073 (w), 1017 (w), 997 (m), 967 (w), 955 (s), 906 (w), 881 (s), 789 (w), 757 (w), 736 (w), 722 (s), 680 (w), 666 (w), 602 (m), 514 (m), 472 (s), 464 (w), 440 (w), 425 (w), 414 (w).

## Modified procedure for the preparation of KC$_8$

A 250 mL round-bottomed Schlenk flask was charged with reagent grade ( > 99.9%) graphite (3.55 g, 818.4 mmol) and dried under dynamic vacuum (1 × 10$^{-3}$ mbar) at 100 °C for 4 h. In an argon-filled glovebox, freshly cut potassium metal (1.45 g, 102.3 mmol) is added. The mixture is then heated under an argon atmosphere with a blowtorch whilst agitating causing the potassium metal to melt and intercalation to occur. Continue heating until the mixture has completely changed colour from black to bronze, which will be for ~3 h. Once the reaction is complete, allow to cool to room temperature. Yield: 5.0 g, 99%.

## Modified procedure for the preparation of UI$_3$(1,4-dioxane)$_{1.5}$

A 500 mL Young's ampoule equipped with a PTFE stirrer bar was charged with uranium turnings (12.18 g, 51.17 mmol). To this, 1,4-dioxane ( ~ 200 mL) was added and the mixture cooled to 0 °C before solid I$_2$ (19.48 g, 76.75 mmol; 1.5 equivalents per U) was added in a portion-wise manner. Note: The addition of I$_2$ is exothermic so I$_2$ should be added slowly to control the reaction. After the addition of I$_2$ was complete, the reaction mixture appeared red in colour. The ampoule was then sealed and the reaction mixture vigorously stirred at room temperature for 7 days. During this time, the red colour dissipated, and a deep purple/blue colour formed with concomitant deposition of solid. After the reaction was complete (no obvious U metal turnings are left), the reaction was concentrated to half volume *in vacuo* and then Et$_2$O (50 mL) added to precipitate the product. The purple/blue solid was collected by filtration through a coarse porosity frit, washed with Et$_2$O (2 × 25 mL), and dried *in vacuo* to afford [UI$_3$(1,4-dioxane)$_{1.5}$] as a purple/blue solid. Yield: 38.05 g, 99%. $^1$H NMR (400 MHz, C$_6$D$_6$, 298 K): δ 3.35 (s, 6H, 1,4-dioxane -C$H_2$) (ppm). ATR-IR v/cm$^{-1}$: 2929 (w), 1449 (m), 1433 (w), 1371 (w), 1297 (s), 1256 (s), 1121 (m), 1093 (s), 1057 (s), 1039 (w), 888 (m), 855 (w), 840 (s), 813 (w), 765 (w), 611 (s).

## Modified procedure for the preparation of [U{N(SiMe$_3$)$_2$}$_3$]

A 250 mL Young's ampoule equipped with a PTFE stirrer bar was charged with a solid mixture of [UI$_3$(1,4-dioxane)$_{1.5}$] (7.50 g, 10 mmol) and [NaN(SiMe$_3$)$_2$] (5.5 g, 30 mmol). Note: Do not use [KN(SiMe$_3$)$_2$] as this will lead to an increased formation of [U(I){N(SiMe$_3$)$_2$}$_3$] as a by-product. The mixture was cooled to −78 °C, then THF (100 mL) was added, and the suspension was allowed to warm slowly to room temperature, during which time there was a colour change from dark blue to dark red/purple. Volatiles were removed *in vacuo* to afford a red/purple residue. Pentane (100 mL) was added and the mixture slurried for 10 min before being filtered through a Celite-padded coarse porosity frit into a 250 mL round bottom Schlenk flask. The resultant filtrate was stored at −30 °C for 24 h, resulting in the precipitation of unwanted NaI and [U(I){N(SiMe$_3$)$_2$}$_3$] by-products. The filtrate is then allowed to warm to room temperature before being filtered to yield a clear red/purple solution. This step was repeated once more, before volatiles were removed *in vacuo* to afford a red/purple residue which was washed with cold (0 °C) SiMe$_4$ (2 × 20 mL). The resultant solid was dried *in vacuo* to afford **1** as a dark purple solid. Yield: 4.50 g, 62.6%. $^1$H NMR (400 MHz, C$_6$D$_6$, 298 K): δ (ppm) −11.31 (s, 54H, Si(C$H_3$)$_3$). ATR-IR v/cm$^{-1}$: 2951 (s), 2897 (w), 1438 (br, w), 1244 (s), 985 (s), 854 (w), 823 (s), 808 (w), 756 (s), 676 (m), 656 (m), 594 (s).

## Modified procedure for the preparation of U$^{IV}$Cl$_4$

A 1000 mL round-bottomed flask was charged with UO$_3$ (23.54 g, 82.22 mmol) and hexachloropropene (250 mL). The flask was equipped with two condensers stacked on top of one another, and the flask placed under an inert gas supply. The mixture was heated carefully to reflux, which was accompanied by a violent exotherm and the liberation of a dark brown gas. The flask was lifted away from the heating mantle to allow the exotherm to subside before heating was resumed. Note: this moderation of the exotherm step may be needed to be conducted multiple times. The reaction mixture was then left to gently reflux for 16 hours. During which time, UCl$_4$ precipitates from solution as a green solid. The mixture was cooled to room temperature, and the reaction mixture was carefully filtered away from the green solid before washing with DCM (3 × 150 mL). Removal of volatiles *in vacuo* afforded UCl$_4$ as a free-flowing green powder, which was used without further purification. Yield: 28.01 g, 90%.

## Modified procedure for the preparation of Tren$^{DMBS}$Li$_3$(THF)$_3$

N(CH$_2$CH$_2$NH$_2$)$_3$ (12 mL, 80.42 mmol) was dissolved in THF (50 mL). $^n$BuLi (2.5 M, 100 mL, 250.00 mmol) was added dropwise at −78 °C, warmed to room temperature and the mixture stirred for 6 h. The solution was then cooled to −78 °C, and ClSiMe$_2$Bu$^t$ (37.6 g, 250.00 mmol) was added in a portion-wise manner and the solution

stirred at room temperature for 16 hours. Removal of volatiles *in vacuo* resulted in a pale-yellow sticky solid. The product was extracted with pentane (2 ×80 mL), and the solution was filtered from the LiCl precipitate. ⁿBuLi (2.5 M, 100 mL, 250.00 mmol) was added dropwise at −78 °C, warmed to room temperature and the solution was stirred for 6 h at room temperature. Removal of volatiles *in vacuo* resulted in an off-white solid which was washed with cold pentane (2 × 20 mL) to yield Tren$^{DMBS}$Li$_3$ as a white powder. Colourless crystals of Tren$^{DMBS}$Li$_3$ were grown from a concentrated solution in hexane stored at −30 °C. Yield: 36.06 g, 62%. $^1$H NMR (400 MHz, C$_6$D$_6$, 298 K): δ (ppm) 0.16 (s, 18H, SiBu$^t$(C$H_3$)$_2$), 1.11 (s, 27H, SiMe$_2$(C(C$H_3$)$_3$)), 1.36 (m, 12H, THF-C$H_2$), 2.37 (t, 6H, NC$H_2$CH$_2$), 3.17 (t, 6H, NCH$_2$C$H_2$), 3.52 (m, 12H, THF-C$H_2$). ATR-IR v/cm$^{-1}$: 2924 (m), 2879 (m), 2846 (m), 2820 (w), 1467 (m), 1384 (w), 1340 (m), 1270 (w), 1236 (s), 1144 (w), 1084 (s), 1058 (m), 1035 (m), 1004 (m), 950 (w), 935 (s), 900 (m), 818 (s), 762 (s), 647 (s), 593 (w), 566 (w), 542 (w), 517 (w), 440 (w), 420 (w).

### Modified procedure for the preparation of [(Tren$^{DMBS}$)U$^{IV}$(Cl)]

A solution of Tren$^{DMBS}$Li$_3$(THF)$_3$ (7.23 g, 10 mmol) in THF (50 mL) was added dropwise to a stirring solution of UCl$_4$ (3.80 g, 10 mmol) in THF (80 mL) at −78 °C. The mixture was allowed to warm to room temperature before stirring for 16 h. Removal of volatiles *in vacuo* resulted in a brown solid. Soluble residues were extracted in hot toluene (100 mL) and the solution was filtered from the LiCl precipitate. Removal of volatiles *in vacuo* resulted in a pale-brown solid which was washed with hexane (2 × 10 mL) to yield [(Tren$^{DMBS}$)U$^{IV}$(Cl)] as a brown solid. Green crystals of [(Tren$^{DMBS}$)U$^{IV}$(Cl)] were grown from a concentrated solution in toluene stored at −30 °C for 24 h. Yield: 6.42 g, 85%. $^1$H NMR (400 MHz, C$_6$D$_6$, 298 K): δ (ppm) −23.36 (s, 6H, NC$H_2$CH$_2$), 6.08 (s, 18H, SiBu$^t$(C$H_3$)$_2$), 6.65 (s, 27H, SiMe$_2$(C(C$H_3$)$_3$)), 7.79 (s, 6H, NCH$_2$C$H_2$). ATR-IR v/cm$^{-1}$: 2949 (s), 2925 (s), 2878 (m), 2851 (s), 1463 (s), 1387 (w), 1359 (w), 1334 (w), 1248 (s), 1140 (w), 1071 (w), 1058 (s), 1021 (m), 1005 (m), 922 (s), 897 (s), 825 (s), 796 (m), 770 (s), 740 (w), 704 (s), 659 (s), 560 (s), 456 (s), 435 (w).

### Modified procedure for the preparation of [(Tren$^{DMBS}$)U$^{IV}$(I)]

A Schlenk flask equipped with a PTFE-coated stirrer bar was charged with solid [(Tren$^{DMBS}$)U$^{IV}$(Cl)] (3.30 g, 4.4 mmol). To this, pentane was added (~ 40 mL) with stirring to form a clear solution to which Me$_3$SiI (2 mL, 14 mmol) was added all at once resulting in a colour change to light brown. The resultant reaction mixture was stirred for 48 h, during which time there was deposition of a brown solid. The suspension was filtered, and the resultant solid was dried *in vacuo* to afford a brown powder, which was washed with pentane (3 × 25 mL). The resultant solid was dried *in vacuo* for one hour to afford [(Tren$^{DMBS}$)U$^{IV}$(I)] as a light brown powder. Yield: 2.32 g, 63%. $^1$H NMR (400 MHz, C$_6$D$_6$, 298 K): δ (ppm) −32.48 (s, 6H, NC$H_2$CH$_2$), 6.32 (s, 6H, NCH$_2$C$H_2$), 9.54 (s, 27H, SiMe$_2$(C(C$H_3$)$_3$)), 11.20 (s, 18H, SiBu$^t$(C$H_3$)$_2$). ATR-IR v/cm$^{-1}$: 2949 (s), 2924 (s), 2879 (m), 2851 (s), 1464 (s), 1387 (w), 1359 (m), 1332 (w), 1252 (s), 1141 (w), 1059 (s), 1021 (m), 922 (s), 896 (s), 825 (s), 797 (m), 773 (s), 739 (m), 698 (s), 659 (s), 564 (s), 459 (s), 440 (w).

### Modified procedure for the preparation of [{[K(Toluene)][(Tren$^{DMBS}$)U$^{Cyclomet}$]}$_2$]

Under an atmosphere of argon, a potassium mirror (20-fold excess relative to U) was formed within a 250 mL Young's ampoule which was then charged with a glass-coated stirrer bar. To this, a slurry of [(Tren$^{DMBS}$)U$^{IV}$(I)] (4.25 g, 5.00 mmol) in toluene (80 mL) was added and the mixture stirred vigorously for 48 hours. The suspension was then filtered, volatiles removed *in vacuo*, and the green solid dried for two hours. Full consumption of [(Tren$^{DMBS}$)U$^{IV}$(I)] was confirmed by $^1$H NMR spectroscopy. The solid was then washed with pentane (2 × 10 mL) and dried *in vacuo* for one hour to yield [{[K(Toluene)][(Tren$^{DMBS}$)U$^{Cyclomet}$]}$_2$] as a green powder. Note: performing this reaction under a dinitrogen atmosphere will result in partial conversion to the

diuranium-N$_2$-complex, [{U$^{IV}$(Tren$^{DMBS}$)}$_2$(μ-η$^2$:η$^2$-N$_2$)][84]. Yield: 2.48 g, 58%. $^1$H NMR (400 MHz, C$_6$D$_6$, 298 K): δ (ppm) −87.96 (s), −54.96 (s), −45.80 (s), −43.41 (s), −38.45 (s), −28.70 (s), −23.39 (s), −6.29 (s), −2.82 (s), 0.88 (s), 1.24 (s), 20.20 (s), 24.79 (s), 26.48 (s), 29.76 (s), 34.61 (s), 35.84 (s), 152.76 (s), 160.11 (s).

### Modified procedure for the preparation of [(Tren$^{DMBS}$)U$^{III}$] (3)

Under an atmosphere of argon, a Schlenk flask was charged with a glass-coated stirrer bar and a solid mixture of [(Tren$^{DMBS}$)U$^{IV}$(Cl)] (0.75 g, 1 mmol) and potassium graphite (0.4 g, 3 mmol, 3 eq.). At −40 °C, hexane (20 mL) was added and the reaction mixture stirred cold before being allowed to warm slowly to room temperature resulting in the formation of a dark brown/purple suspension, which was stirred for a further 36 h. The suspension was then filtered, volatiles removed *in vacuo*, and the dark purple solid dried for 2 h. Full conversion to the [(Tren$^{DMBS}$)U$^{III}$] species was confirmed by $^1$H NMR spectroscopy. Under an atmosphere of argon, pentane (2 mL) was added to form a very dark purple solution which was stored at −30 °C for 3 days to yield dark-purple crystals of [(Tren$^{DMBS}$)U$^{III}$]. Note: performing this reaction under a dinitrogen atmosphere will result in partial conversion to the diuranium-N$_2$-complex, [{U$^{IV}$(Tren$^{DMBS}$)}$_2$(μ-η$^2$:η$^2$-N$_2$)][84]. Yield: 0.25 g, 65%. $^1$H NMR (400 MHz, C$_6$D$_6$, 298 K): δ (ppm) −37.01 (s, 6H, NC$H_2$CH$_2$), −1.55 (s, 18H, SiBu$^t$(C$H_3$)$_2$), 9.81 (s, 27H, SiMe$_2$(C(C$H_3$)$_3$)), 21.64 (s, 6H, NCH$_2$C$H_2$). ATR-IR v/cm$^{-1}$: 2949 (w), 2923 (m), 2878 (m), 2849 (w), 2830 (m), 1462 (s), 1441 (w), 1386 (w), 1357 (w), 1344 (w), 1247 (s), 1122 (m), 1097 (w), 1061 (s), 1021 (w), 1004 (w), 924 (s), 822 (w), 799 (s), 769 (s), 737 (w), 700 (m), 660 (w), 648 (m), 585 (w), 555 (s), 511 (w), 442 (s).

### Preparation of [(Tren$^{DMBS}$)U$^{IV}$(NMe$_2$)] (4)

At −78 °C, a pale-yellow solution of **1** (0.13 g, 0.5 mmol) in toluene (5 mL) was added to a dark red purple solution of **3** (0.36 g, 0.5 mmol) in toluene (10 mL), and then the mixture was slowly warmed to room temperature. After being stirred for 24 h, the mixture turned into a dark red brown solution, and removal of volatiles *in vacuo* gave a brown solid residue. Pentane (5 mL) was added to the residue to afford a white slurry which was stored at −30 °C for 6 h to ensure all the anthracene and unreacted **1** precipitated out of the red brown solution. After filtration, the filtrate was concentrated to ~2 mL and stored at −30 °C for 24 h, giving **4** as dark brown crystals. The crystalline solid was isolated by decanting the mother liquor before being washed with cold SiMe$_4$ (2 × 1 mL), and then dried *in vacuo*. Yield: 0.13 g, 32%. $^1$H NMR of the crude product indicated the amide species **4** was the main product, and there was no sign of a diuranium-diphosphorus complex; this might be due to the diphosphorus ligand being too reactive to be stabilized by the less sterically demanding Tren$^{DMBS}$ ligand environment. Moreover, by $^{31}$P NMR spectroscopy there was no evidence for P$_4$ formation in the crude products, so the fate of PDBN P atom remains unknown. In addition, **3** does not react with PDBN-P(BPh$_3$) Na(OEt$_2$)$_2$ or PDBN-PBu$^t$, reflecting the more reactive nature of PDBNP-H/-NMe$_2$ as phosphinidene group transfer reagents. It should be noted that **4** can also be prepared by the reaction of [(Tren$^{DMBS}$)U$^{IV}$(I)][82] with LiNMe$_2$ via a salt metathesis method: At −78 °C, a colourless solution of LiNMe$_2$ (0.025 g, 0.5 mmol) in THF (5 mL) was added to a brown solution of [(Tren$^{DMBS}$)U$^{IV}$(I)] (0.42 g, 0.5 mmol) in THF (20 mL), and then the mixture was slowly warmed up to room temperature. After being stirred for 24 h, the mixture turned into a brown solution, and removal of volatiles *in vacuo* gave a brown solid residue. Pentane (15 mL) was added to the residue to afford a brown slurry which was filtered, and the filtrate was concentrated to ~4 mL and stored at −30 °C for 24 h, yielding **4** as dark brown crystals. The crystalline solid was isolated by decanting the mother liquor before being dried *in vacuo*. Yield: 0.25 g, 65%. Anal. Calcd for C$_{26}$H$_{63}$N$_5$Si$_3$U: C, 40.66; H, 8.27; N, 9.12%. Found: C, 40.57; H, 8.12; N, 9.25%. $^1$H NMR (400 MHz, C$_6$D$_6$, 298 K): δ (ppm) −24.48 (s, 18H, SiBu$^t$(C$H_3$)$_2$), −11.79 (s, 27H,

SiMe$_2$(C(CH$_3$)$_3$), 8.55 (s, 6H, N(CH$_3$)$_2$), 59.01 (s, 6H, NCH$_2$CH$_2$), 100.61 (s, 6H, NCH$_2$CH$_2$). $^{29}$Si{$^1$H} NMR (79 MHz, C$_6$D$_6$, 298 K): δ (ppm) − 192.59 (s). ATR-IR v/cm$^{-1}$: 2950 (m), 2922 (m), 2849 (m), 2816 (m), 2761 (w), 1460 (m), 1384 (w), 1356 (w), 1331 (w), 1248 (m), 1136 (w), 1056 (m), 1024 (w), 923 (s), 891 (m), 820 (s), 767 (s), 706 (s), 653 (m), 566 (w), 493 (m), 452 (w).

### Modified procedure for the preparation of [(Ar*O)$_3$U$^{III}$] (5)

A solution of [U{N(SiMe$_3$)$_2$}$_3$] (3.60 g, 5.00 mmol) in hexane (20 mL) was added dropwise to a stirring solution of Ar*OH (3.10 g, 15.00 mmol, Ar* = 2,6-$^t$Bu$_2$C$_6$H$_3$) in hexane (20 mL) at −78 °C. The mixture was allowed to warm to room temperature before stirring for 16 hours, resulting in the formation of a dark green/black suspension. The suspension was then filtered and volatiles removed in vacuo, resulting in a dark green solid, which was washed with hexane (2 × 5 mL) to yield 5 as a dark green/black solid. Yield: 3.08 g, 72%. $^1$H NMR (400 MHz, C$_6$D$_6$, 298 K): δ (ppm) −6.09 (s, 54 H, Ar-$^t$Bu), 13.69 (s, 3H, para-Ar-H), 16.61 (s, 6H, meta-Ar-H). ATR-IR v/cm$^{-1}$: 3065 (w), 2951 (br, m), 2908 (w), 1582 (m), 1457 (m), 1407 (s), 1385 (m), 1354 (m), 1263 (w), 1232 (s), 1194 (m), 1154 (w), 1123 (m), 1097 (s), 922 (w), 882 (w), 862 (s), 817 (s), 797 (w), 746 (s), 653 (s), 594 (m), 545 (s), 451 (s).

### Preparation of [{(Ar*O)$_3$U$^{IV}$}$_2$(μ-η$^2$:η$^2$-P$_2$)] (6)

At −78 °C, toluene (30 mL) was added to the solid mixture of 5 (0.43 g, 1.00 mmol) and 1 (0.13 g, 0.50 mmol), and then the mixture was warmed to room temperature. After being stirred for 24 h at room temperature, the mixture turned into a yellow brown solution, from which volatiles were removed in vacuo to afford a brown solid residue. The anthracene by-product was removed by sublimation (80 °C, 2.0 × 10$^{-6}$ mbar) and the remaining residue extracted with pentane (5 mL) and filtered, resulting in a brown filtrate which was concentrated to ~3 mL. Storing this brown filtrate at −30 °C for 24 h produced red crystals of 6 suitable for single-crystal X-ray diffraction studies, but the crystalline samples were mixed with a brown solid (putatively assigned as [(Ar*O)$_3$U$^{IV}$(NMe$_2$)] product) which could not be separated due to each having essentially identical solubilities in common solvents such as pentane, Et$_2$O, and toluene. Attempts to manually separate the two solids were unsuccessful due to the small size of the crystals.

### Preparation of [{(Tren$^{DMBS}$)U$^{IV}$}$_2$(μ-PH)] (7) and [{(Tren$^{DMBS}$)U$^{IV}$}$_2${μ-η$^2$:η$^2$-(HPPH)}] (8)

At −78 °C, a dark purple solution of 3 (0.73 g, 1.00 mmol) in toluene (10 mL) was added to a colourless solution of 2 (0.23 g, 1.10 mmol) in toluene (10 mL), and then the mixture was warmed up to room temperature. After being stirred for 30 min at room temperature, the mixture turned into a bright red solution, and removal of volatiles in vacuo gave a dark red solid residue. The anthracene by-product was removed by sublimation (80 °C, 2.0 × 10$^{-6}$ mbar) and the remaining residue was extracted with pentane (5 mL) and filtered, resulting a bright red filtrate which was concentrated to ~2 mL and stored at −30 °C for 24 h, yielding 8 as dark red crystals that were suitable for a single-crystal X-ray diffraction studies. The crystalline solid was isolated by decanting the mother liquor, washing with cold SiMe$_4$ (2 × 1 mL), and then drying in vacuo. Yield: 0.18 g, 36% (by U content, maximum 66% yield as the $^1$H NMR of the crude product showed that the ratio of 8:7 = 2:1). Note: adding a solution of 2 in toluene to that of 3 in toluene does not affect the reaction outcome and extending the reaction time longer than 2 hours leads to the decomposition of the diphosphene product resulting in a lower yield. Anal. Calcd for C$_{48}$H$_{116}$N$_8$P$_2$Si$_6$U$_2$(pentane): C, 38.13; H, 7.73; N, 7.41%. Found: C, 37.96; H, 7.82; N, 7.34%. $^1$H NMR (400 MHz, C$_6$D$_6$, 298 K): δ (ppm) −147.57 (s, 2H, HPPH), −17.93 (s, 12H, NCH$_2$CH$_2$), −4.45 to −3.10 (s, br, 36H, SiBu$^t$(CH$_3$)$_2$), 2.40 (s, 12H, NCH$_2$CH$_2$), 5.86 (s, 54H, SiMe$_2$(C(CH$_3$)$_3$). $^{31}$P NMR (162 MHz, C$_6$D$_6$, 298 K): δ (ppm) 1065.44 (s, br, due to its broad

nature, the P-H coupling was not observed). $^{29}$Si{$^1$H} NMR (79 MHz, C$_6$D$_6$, 298 K): δ (ppm) −42.43 (s). ATR-IR v/cm$^{-1}$: 2952 (m), 2924 (m), 2878 (m), 2847 (m), 2248 (P-H stretch, w), 1469 (m), 1387 (w), 1359 (w), 1331 (w), 1244 (m), 1145 (w), 1049 (s), 1023 (m), 925 (s), 891 (m), 825 (s), 770 (s), 709 (s), 656 (s), 570 (m), 450 (m). Raman v/cm$^{-1}$: 2895 (w), 2842(w), 1465 (w), 1238 (w), 841 (w), 586 (w), 567 (w), 448 (s). Complex 7 was isolated from the reaction mixture by the following work-up: the above insoluble residue was extracted with THF (2 mL) and filtered, affording a dark red brown filtrate which was stored at −30 °C for 2 days, yielding a small crop of dark red brown crystals of 7. The crystalline solid was isolated by decanting the mother liquor, washing with pentane (2 × 1 mL), and then drying in vacuo. Yield: 0.03 g, 12% (by U content, maximum 34% yield). It should also be noted that 7 can also be prepared by the reaction of a known cyclometallated U$^{III}$ species [{[K(toluene)][(Tren$^{DMBS}$)U$^{Cyclomet}$]}$_2$] with 2. At −78 °C, toluene (20 mL) was added to the pre-cooled solid mixture of [{[K(toluene)][(Tren$^{DMBS}$) U$^{Cyclomet}$]}$_2$] (0.86 g, 0.50 mmol) and 2 (0.11 g, 0.50 mmol) and the mixture was allowed to warm up to room temperature. After being stirred for 24 hours, the mixture turned into a dark red solution. The reaction was filtered, and the dark filtrate was concentrated to ~5 mL and stored at −30 °C for 24 h, yielding 7 as dark brown crystals. The crystalline solid was isolated by decanting the mother liquor and drying in vacuo. Yield: 0.31 g, 42%. Anal. Calcd for C$_{48}$H$_{115}$N$_8$PSi$_6$U$_2$: C, 38.95; H, 7.83; N, 7.57%. Found: C, 39.00; H, 7.93; N, 7.49%. $^1$H NMR (400 MHz, C$_6$D$_6$, 298 K): δ (ppm) −154.14 (s, 1H, PH), −5.03 (s, 12H, NCH$_2$CH$_2$), −1.35 (s, 36H, SiBu$^t$(CH$_3$)$_2$), 2.87 (s, 54H, SiMe$_2$(C(CH$_3$)$_3$, 5.88 (s, br, 12H, NCH$_2$CH$_2$). $^{31}$P NMR (162 MHz, C$_6$D$_6$, 298 K): δ (ppm) not observed due to its broad nature and the poor solubility of the complex in benzene or THF. $^{29}$Si{$^1$H} NMR (79 MHz, C$_6$D$_6$, 298 K): δ (ppm) −85.47 (s). ATR-IR v/cm$^{-1}$: 2951 (m), 2924 (m), 2875 (m), 2841 (m), 2185 (br, PH), 1468 (m), 1445 (w), 1407 (w), 1388 (w), 1331 (w), 1247 (m), 1147 (w), 1058 (s), 1028 (w), 923 (s), 894 (m), 817 (s), 798 (s), 783 (s), 739 (s), 707 (s), 662 (s), 562 (m), 455 (s). Reliable UV/Vis/NIR spectra could not be obtained as dinuclear 7 is poorly soluble in common solvents such as benzene, toluene or THF.

### Preparation of [{(Ar*O)$_3$U$^{IV}$}$_2${μ-η$^2$:η$^2$-(HPPH)}] (9)

At −78 °C, toluene (30 mL) was added to the solid mixture of 5 (0.85 g, 1.00 mmol) and 2 (0.23 g, 1.10 mmol), and the mixture warmed up to room temperature with stirring. After 2 h the mixture turned into a red solution, and removal of volatiles in vacuo gave a red solid residue. The by-product anthracene was removed by sublimation (80 °C, 2.0 × 10$^{-6}$ mbar) and the residue was extracted with toluene (10 mL) and filtered, resulting in a bright red filtrate which was concentrated to ~3 mL. Pentane ( ~ 3 mL) was layered on top of the filtrate and this was stored at −30 °C for 24 h, yielding 9 as dark red crystals that were suitable for single-crystal X-ray diffraction studies. The crystalline solid was isolated by decanting the mother liquor, washing with pentane (2 × 1 mL), and then drying in vacuo. Yield: 0.42 g, 47%. Anal. Calcd for C$_{84}$H$_{128}$O$_6$P$_2$U$_2$: C, 56.94; H, 7.28; N, 0%. Found: C, 56.60; H, 7.24; N, 0%. $^1$H NMR (400 MHz, C$_6$D$_6$, 298 K): δ (ppm) −175.99 (s, 2H, HPPH), −12.95 (s, 94H, Ar-$^t$Bu), −12.08 (s, 14H, Ar-$^t$Bu), 16.98 (s, 6H, para-Ar-H), 20.63 (s, 12H, meta-Ar-H). $^{31}$P NMR (162 MHz, C$_6$D$_6$, 298 K): δ (ppm) 844.40 (s, due to its broad nature, the P-H coupling was not observed). ATR-IR v/cm$^{-1}$: 2954 (m), 2917 (m), 2864 (m), 2265 (P-H stretch, w), 1581 (w), 1459 (w), 1400 (s), 1359 (w), 1255 (w), 1179 (vs), 1111 (s), 930 (w), 852 (vs), 817 (s), 743 (s), 655 (s), 544 (w), 448 (w).

### General computational details

Calculations on 6, 7, 8, and 9 were performed using coordinates derived from their respective crystal structures as the starting points. No constraints were imposed on the structures during the geometry optimizations. The calculations were performed using the Amsterdam Density Functional (ADF) suite version 2017 with standard convergence criteria[91,92]. The DFT geometry optimizations employed

Slater type orbital (STO) triple-ζ-plus polarization all-electron basis sets (from the Dirac and ZORA/TZP database of the ADF suite). Scalar relativistic approaches (spin-orbit neglected) were used within the ZORA Hamiltonian[93–95] for the inclusion of relativistic effects and the local density approximation (LDA) with the correlation potential due to Vosko *et al* was used in all of the calculations[96]. Generalized gradient approximation (GGA) corrections were performed using the functionals of Becke and Perdew[97,98]. Analytical frequency calculations were carried out within the ADF program. The Quantum Theory of Atoms in Molecules analysis[99,100] was carried out within the ADF program. The ADF-GUI (ADFview) was used to prepare the three-dimensional plots of the electron density.

### Computational reaction profile details

The optimization of three different spin states for uranium complexes were carried out by employing DFT hybrid functional (B3PW91)[101,102] along with small core pseudopotential Stuttgart basis set for uranium, phosphorus atoms with additional polarization functions for phosphorus atoms[103–107]. Pople basis sets (6-31 G**) were employed for the rest of the atoms[108,109]. Frequency calculations were performed to locate minima for the optimized structures. All the calculations were performed using Gaussian 16 suite of programs[110]. Dispersion corrections were included in our calculations by employing D3 version of Grimme's dispersion with Becke-Johnson damping[111].

## Data availability

Crystallographic data for the structures reported in this Article have been deposited at the Cambridge Crystallographic Data Centre, under deposition numbers CCDC 2424900 (**4**), 2424901 (**6**), 2424902 (**7**), 2424903 (**8**), and 2424904 (**9**). Copies of the data can be obtained free of charge via https://www.ccdc.cam.ac.uk/structures/. All the other data supporting the findings of this study are available within the Article or its Supplementary Information (Supplementary Figs. 1–64 and Tables 1–6). DFT geometry optimized Cartesian coordinates source data are provided with this manuscript. The spectroscopic and magnetic source data in this study have been deposited in the Figshare database [https://doi.org/10.48420/30178576][112]. Source data are provided with this paper.

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

## Acknowledgements

We gratefully acknowledge funding and support from the UK Engineering and Physical Sciences Research Council (grants EP/K024000/1, EP/M027015/1, EP/P001386/1, EP/S033181/1, and EP/T011289/1), European Research Council (grant CoG612724), Royal Society (grant UF110005) and The University of Manchester including computational resources and associated support services from the Computational Shared Facility. The Alexander von Humboldt Foundation is thanked for a Friedrich Wilhelm Bessel Research Award to S.T.L. L.M. is a senior member of the Institut Universitaire de France. CalMip is acknowledged for a generous grant of computing time. We thank Martin Jennings and Anne Davies at the University of Manchester for acquiring CHN microanalyses.

## Author contributions

J.D. synthesized and characterized all the compounds. T.R. conducted and analyzed the reaction profile calculations. J.A.S. recorded and interpreted the magnetic data. A.J.W. collected and refined the crystallographic characterization. J.D. and S.T.L. conceived the original idea. L.M. and S.T.L. supervised the work, analyzed all the data and wrote the manuscript with contributions from all the authors.

## Competing interests

The authors declare no conflict of interest.
