## [Transparent Peer Review file · Nature Communications]

Isolable f-Element Diphosphene Complexes by Phosphinidene Group Transfer and Coupling at Uranium

Corresponding Author: Professor Stephen Liddle

Version 0:

Reviewer comments:

Reviewer #1

(Remarks to the Author)

The present paper by Maron and Liddle, discusses the synthesis and characterization of hitherto unknown f-element (uranium) diphosphene complexes and their in-depth characterization using various spectroscopic and computational techniques. The synthetic, spectroscopic and theoretical work is conducted on an extremely high and sophisticated level and deliver valuable insights into a missing link in P2 chemistry, not only for f-element chemists, but also for a large readership of (inorganic) chemists. Thus, I highly recommend this article for publication, once some minor issues have been addressed:

In the introduction, the authors describe the complexes I – IV, however, I personally think that for a broad audience, unfamiliar with previous work of the Liddle Group, it would be highly beneficial to also add a small scheme / figure, representing them (asides from their description in the text).

In Figure 2, two things could be optimized. First of all, it seems like that part a described in the caption got lost and no reactions with “prior” results are shown. Second, I would suggest to replace the abbreviation Ar* with just A rot tBuAr or something like that. The reason I am saying this is, that Ar* often is associated with Mes* (tri-tert-butylphenol) or terphenyl substituents. Furthermore, I think it would be beneficial to have the schematic ChemDraw representation of 2,6 di-tert-butylphenol? Either as a side picture or one of the Ar* groups could be fully shown, while the others are abbreviated in the structure of 6 and 9?

Do the authors have any explanation, why 9 is formed selectively, while 8 is “contaminated” with 7? Is it possible that in the reaction leading to 9 a related diuranium complex is formed, but somewhere lost during the workup? Does the crude ³¹P NMR of 9 show only one species?

Figures 3 and 4 could be combined as they both show the “integral” structures of the manuscript. To me it makes no sense, to have them appear as two different figures at different discussion points in the manuscript.

The following is a long shot, and I know that this is often not observable for paramagnetic complexes in the NMR, but given sensitivity of ¹H and ³¹P NMR spectroscopy: have the authors ever seen a ¹J_{PH} coupling in either of the complexes? Although the presence of the H atom is validated by IR and X-ray, I feel that seeing / finding the ¹J_{PH} coupling in any of the NMR experiments would be highly beneficial for the manuscript. If 1D measurements do not work, to the best of my knowledge ¹H ³¹P HSQC measurements are another tool to unambiguously proof the presence of a P₂H₂ ligand.

To finally add a question regarding the chemistry of the complexes 8 and 9, can they be further deprotonated to yield a putative P₂ 4- unit? I personally think, that the arene groups in 9 would be highly beneficial to also “trap” any counterions, stabilizing such a highly charged bridge.

And to further investigate the chemical stability of the [P₂H₂]²⁻ bridge. Is it possible to transform 9 into 6 through the addition of 2,4,6-tri-tert-butylphenoxy radicals as H(dot) abstractors?

Finally, I would like to congratulate all authors again to a really fantastic piece of chemistry and congratulate all authors again to these nice results and fine manuscript, which I really enjoyed reading and reviewing.

Kind regards,
Stephan Hohloch

Reviewer #2

(Remarks to the Author)

The manuscript from Liddle and co-workers describes the synthesis of two novel [HPPH] containing compounds, in which the formal dihydrodiphosphene ligand bridges two U centres, and is best described as the dianionic form. These species are formed through the addition of phospho-dibenzonorbornadienes, known to be effective P(I)-transfer reagents, to U(III) synthons; this ultimately leads to amide transfer or P(I) transfer, the latter forming coordinated P2 or the title P2H2 units. The formation of these different species is justified by DFT calculations. The work has been carried out to a high standard, with nice experimental work, considering that this is likely quite fiddly chemistry. Nonetheless, I see this as highly specialized work, which is iterative to earlier described U-P species accessed using what are essentially the same methodologies, with no true breakthrough given the similarities in synthetic approach, e.g. in forming P2 species. I would as such recommend submission to a more specialised organometallic journal. Still, I commend the authors on some interesting compounds. I also have the following comments, which should be addressed before further submission:

- o Line 6 of intro: I'm not quite sure what the authors mean by 'diphosphinyldene'? Perhaps phosphaphosphanylidene would be better used to describe H2P-P.
- o Line 7-8 of intro: "...stabilization of HPPH in the condensed phase by metal centres is in principle a practical way to stabilize and..." should be re-read and re-phrased (e.g. coordination of HPPH by metal centres).
- o Line 15 of intro: Figure 1 mid-sentence, not in parentheses, should be changed (i.e. in parentheses, end of sentence).
- o Figure 1 caption: there are two commas after '34' which I assume should be there.
- o Generally in the introduction, I find the use and placement of references confusing. It seems the reference is placed in some cases where the referenced compound is mentioned, sometimes when a work is referred to, and sometimes elsewhere. The first sentence of page 3, for example, has 6 different points of placement for references. I would personally avoid referencing starting materials here, to make it clear which references refer to which statement (made by the sentence, not a bracketed compound).
- o Following from the above, reference 34 describes a thorium-antimony complex. It's unclear what relevance this has in the present discussion of U-P species.
- o Again related to the above, the authors may consider clarifying some points by shortening sentences (the above example is a 7-liner).
- o Page 3, line 7: the – after P should presumably be superscript.
- o Figure 3: the caption describes parts (a) and (b), but only the species described in part (b) are shown. Not sure if the caption or the figure are incorrect here. Please change.
- o On the syntheses, the authors suggest the synthetic route is a promising pathway to access HPPH species, but they mention in almost all cases the multiple products form, which in some cases are inseparable (e.g. in the synthesis of 6). This, in my view, detracts from the utility. The authors might consider rephrasing such points.
- o The X-ray data for compound 8 are not of a publishable standard, with multiple peaks of residual density of between 2 and 4(!). The authors state that this is due to the heavy atom; if this was not fixed by an absorption correction, it is probably due to poor data quality (e.g. not a single crystal). This data should be recollected, and should not be published as it stands – such Q-peaks will greatly affect the accuracy of discussed metrical parameters in the core of the structure, especially given that these peaks are in the vicinity of both P and U.

Reviewer #3

(Remarks to the Author)

Review of "Isolable f-Element Diphosphene Complexes by Phosphinidene Group Transfer and Coupling at Uranium"
Synopsis: The manuscript reports isolation of (HPPH)₂- stabilized by U(IV) centers, adding to f-element HNNH and HASAsH complexes reported before. Combining experiments and detailed computational studies, the authors conclude reaction outcomes are controlled by the ancillary ligands, the radical nature of intermediates, and the phosphinidene substituent. They further propose the unfavourability of U adopting high (V/VI) oxidation states to form multiple polar covalent bonds to P.
Recommendation: Publish on a more specialized journal.

Since structurally relevant N, As, and P compounds have previously been reported, the synthetic novelty of the current study is limited. Probably the most important insight mentioned in this manuscript is "By contrast, the analogous one-electron oxidation of [UIV(TrenTIPS)(PH)]⁻ results in the isolation of [UIV(TrenTIPS)(PH₂)] and II,49" Unfortunately, this is published

in a previous work by the same group of authors. Overall, the authors have worked on this system for a long time and published a lot of works already. This work seems to be a mere extension of previous work. Therefore, it may suit a more specialized journal.

Major comments for improving the manuscript:

1. A Scheme showing relevant reactivity for As should be included in addition to Figure 1, so it will be easier for the readers to compare P and As.
2. Reaction conditions (temperature, solvents, duration) and yields should be added in Figure 2.
3. No clear shoulder can be seen for compound 8 in the xT plot.
4. "The data are similar to those of $[(\text{TrenTIPS})\text{U}(\text{IV})_2\{\mu\text{-h}_2\text{h}_2\text{-}(\text{HAsAsH})\}]$ (MDCq U/As = 3.20/-1.12),²³ but overall are less polarized for 8 and certainly 9, suggesting that HPPH is a more effective donor to U than HAsAsH, or put another way HAsAsH is a better acceptor than HPPH, as predicted theoretically." This part is confusing. The charge on P is slightly more negative than that of As (-1.14 vs. -1.12). How the authors come to the conclusion that HAsAsH is better acceptor than HPPH? Same for the argument about the comparison between 8 and 9. The difference is subtle and purely DFT-based. The authors should restrain themselves from too many speculations.
4. Some quantitative analysis should be done to verify the statement about the steric hindrance of different ancillary ligands. It is not obvious that three Ar⁺O ligands are less sterically demanding than Tren ligands. In addition, this reviewer doesn't understand why (P2)²⁻ can be isolated for TrenTIPS but not (HPPH)²⁻. The U-P distance should be longer for (HPPH)²⁻ than for (P2)²⁻, so the (HPPH)²⁻ bridging species should have very similar or slightly less

Minor comments:

1. For ref.20, it's a "Ge" complex, not "Ga". The typo showed up in page 2 and 8.
2. Reaction of P4 and molybdenum hydride precursors was firstly reported by Green and co-workers in 1974, while until 1977 the structure was confirmed by Cannillo et al., please add "J. Chem. Soc., Chem. Commun. 1974, 212." to ref.18.
3. Figure 2 seems missing part "a) prior work showing the reaction of I with 1 to give the diphosphorus complex II and with 2 to give the phosphinidene complex III."
4. There's a typo in ref.21, the page number should be "1081-1082" instead of "1981-1982".

Reviewer #4

(Remarks to the Author)

Version 1:

Reviewer comments:

Reviewer #1

(Remarks to the Author)

Liddle and co-workers have thoroughly addressed all of the reviewers' comments and clearly addressed the criticisms raised by the reviewer. Additionally, they have revised the manuscript in accordance with the reviewers' suggestions, which, in my view, has significantly improved the quality and readability of the manuscript.

I agree with the group's argument that this work is not merely an extension of their previous research, but rather fills an important gap in phosphorus and f-element chemistry through the successful formation of an HPPH²⁻ ligand. Therefore, I believe the work will strongly appeal to the readers of Nature Communications and is likely to inspire many chemists to further explore f-element chemistry using phosphinidene transfer reagents

REVIEWER COMMENTS

Reviewer #1 (Remarks to the Author):

The present paper by Maron and Liddle, discusses the synthesis and characterization of hitherto unknown f-element (uranium) diphosphene complexes and their in-depth characterization using various spectroscopic and computational techniques. The synthetic, spectroscopic, and theoretical work is conducted on an extremely high and sophisticated level and deliver valuable insights into a missing link in P2 chemistry, not only for f-element chemists, but also for a large readership of (inorganic) chemists. Thus, I highly recommend this article for publication, once some minor issues have been addressed:

RESPONSE: We thank the reviewer for their support and thoughts and address their queries below.

1. In the introduction, the authors describe the complexes I – IV, however, I personally think that for a broad audience, unfamiliar with previous work of the Liddle Group, it would be highly beneficial to also add a small scheme / figure, representing them (asides from their description in the text).

RESPONSE: That is an excellent suggestion, and we agree. Therefore we have added a 'prior work' pane to Figure 2.

In Figure 2, two things could be optimized. First of all, it seems like that part a described in the caption got lost and no reactions with "prior" results are shown. Second, I would suggest to replace the abbreviation Ar* with just A rot tBuAr or something like that. The reason I am saying this is, that Ar* often is associated with Mes* (tri-tert-butylphenol) or terphenyl substituents. Furthermore, I think it would be beneficial to have the schematic ChemDraw representation of 2,6 di-tert-butylphenol? Either as a side picture or one of the Ar* groups could be fully shown, while the others are abbreviated in the structure of 6 and 9?

RESPONSE: The reviewer is correct, there was a mix-up with the scheme that is now resolved by addressing the above comment. In terms of Ar* vs Mes*, we think there are quite a few uses in the literature, but we personally regard Mes* as always being super-Mes and we are keen to avoid lengthy formula-type representations so feel that since it is defined in the [context of the] paper Ar* is fine. But we agree about showing what Ar* is more clearly since the formula is there but easily missed so now we have added a picture of Ar* to the bottom of the figure.

Do the authors have any explanation, why 9 is formed selectively, while 8 is "contaminated" with 7? Is it possible that in the reaction leading to 9 a related diuranium complex is formed, but somewhere lost during the workup? Does the crude ³¹P NMR of 9 show only one species?

RESPONSE: The reviewer raises a good point. When we investigated the reaction of [(Ar*O)₃U] (5) with Anth-PH (1) the crude ¹H/³¹P NMR show that the reaction is very clean, with only 9 observed along with anthracene and some free proligand, so it would seem that no other species such as U(PH)U was produced. We already commented on why this is likely the case in the synthesis and discussion sections but have now also added a comment about the NMR of the crude mixture into the section describing the synthesis of 9.

Figures 3 and 4 could be combined as they both show the "integral" structures of the manuscript. To me it makes no sense, to have them appear as two different figures at different discussion points in the manuscript.

RESPONSE: Fair point, we have merged the two key XRD structures into one figure and renumbered the figures accordingly.

The following is a long shot, and I know that this is often not observable for paramagnetic complexes in the NMR, but given sensitivity of ¹H and ³¹P NMR spectroscopy: have the authors ever seen a 1JPH coupling in either of the complexes? Although the presence of the H atom is validated by IR and X-ray, I feel that seeing / finding the 1J PH coupling in any of the NMR experiments would be highly beneficial for the manuscript. If 1D measurements do not work, to the best of my knowledge ¹H ³¹P HSQC measurements are another tool to unambiguously proof the presence of a P2H2 ligand.

RESPONSE: The reviewer raises an interesting point. We did look at this before submission, but, unfortunately, we did not observe any coupling and there is no obvious change to the broad resonances running ^{31}P coupled or decoupled. This is a typical issue for paramagnetic compounds when the relevant nuclei are in such close proximity to paramagnetic centres and so is not unexpected and also affects HSQC. However, as the reviewer mentions other methods validate the formulations.

To finally add a question regarding the chemistry of the complexes 8 and 9, can they be further deprotonated to yield a putative P_2^{4-} unit? I personally think, that the arene groups in 9 would be highly beneficial to also “trap” any counterions, stabilizing such a highly charged bridge. And to further investigate the chemical stability of the $[\text{P}_2\text{H}_2]^{2-}$ bridge. Is it possible to transform 9 into 6 through the addition of 2,4,6-tri-tert-butylphenoxy radicals as $\text{H}(\cdot)$ abstractors?

RESPONSE: We thank the reviewer for their suggestions, and we also considered these matters before submission since we had the same thought about favourable $\text{K} \dots \text{arene}$ interactions. However, thus far, sadly, we seem to observe decomposition. Those are negative results, and hence not definitive so we are reluctant to conclude that is end of the matter, and we think it wouldn't be reasonable to delay publication for the possibility of an eventual success so we have decided to not include those reactions in the manuscript.

Finally, I would like to congratulate all authors again to a really fantastic piece of chemistry and congratulate all authors again to these nice results and fine manuscript, which I really enjoyed reading and reviewing.

RESPONSE: We thank the reviewer for their support and thoughts.

Kind regards,
Stephan Hohloch

Reviewer #2 (Remarks to the Author):

The manuscript from Liddle and co-workers describes the synthesis of two novel [HPPH] containing compounds, in which the formal dihydrodiphosphene ligand bridges two U centres, and is best described as the dianionic form. These species are formed through the addition of phosphadibenzonorbondienes, known to be effective $\text{P}(\text{I})$ -transfer reagents, to $\text{U}(\text{III})$ synthons; this ultimately leads to amide transfer or $\text{P}(\text{I})$ transfer, the latter forming coordinated P_2 or the title P_2H_2 units. The formation of these different species is justified by DFT calculations. The work has been carried out to a high standard, with nice experimental work, considering that this is likely quite fiddly chemistry. Nonetheless, I see this as highly specialized work, which is iterative to earlier described U-P species accessed using what are essentially the same methodologies, with no true breakthrough given the similarities in synthetic approach, e.g. in forming P_2 species. I would as such recommend submission to a more specialised organometallic journal. Still, I commend the authors on some interesting compounds. I also have the following comments, which should be addressed before further submission:

RESPONSE: We thank the reviewer for their thoughts and address their queries below. However, we respectfully disagree with the opinion that this is specialised work because: (i) molecular complexes containing HPPH remain exceedingly rare across the whole Periodic Table; (ii) these are the first f-element derivatives filling the gap created by reports of f-element complexes of HNNH and HAsAsH in 1992 and 2015, respectively; (iii) these HPPH linkages are made by a new methodology for HPPH; (iv) the work informs us about how these phosphinidene transfer reagents operate generally teasing out common underlying key reaction steps and then divergences and their reasons for occurring; (v) the work throws up the surprise result of the stability of the open shell intermediates vs the anticipated electron paired formulations.

1. Line 6 of intro: I'm not quite sure what the authors mean by 'diphosphyldene'? Perhaps phosphaphosphanylidene would be better used to describe $\text{H}_2\text{P}-\text{P}$.

RESPONSE: The reviewer raises an interesting question, though that is how H_2PP is referred to in reference 8. P-nomenclature is never straightforward, and we see where the reviewer is coming from with their suggestion, but to us diphosphyldene seems to be more concise and is also consistent with diphosphene so we prefer to keep it as it is.

2. Line 7-8 of intro: "...stabilization of HPPH in the condensed phase by metal centres is in principle a practical way to stabilize and..." should be re-read and re-phrased (e.g. coordination of HPPH by metal centres).

RESPONSE: We thank the reviewer for the suggestion, we realised that there were two instances of stabilise in the sentence. We have thus changed the second instance to isolate.

3. Line 15 of intro: Figure 1 mid-sentence, not in parentheses, should be changed (i.e. in parentheses, end of sentence).

RESPONSE: We thank the reviewer for highlighting the text. The positioning was correct and ,, or () is an editorial house-style matter, but we agree with the underlying notion that the sentence is better split up which we have done.

4. Figure 1 caption: there are two commas after '34' which I assume should be there.

RESPONSE: Good spot, this has been corrected.

5. Generally in the introduction, I find the use and placement of references confusing. It seems the reference is placed in some cases where the referenced compound is mentioned, sometimes when a work is referred to, and sometimes elsewhere. The first sentence of page 3, for example, has 6 different points of placement for references. I would personally avoid referencing starting materials here, to make it clear which references refer to which statement (made by the sentence, not a bracketed compound).

RESPONSE: We are a bit puzzled by this comment because references are placed in the normal way, though sometimes to avoid confusion with numbers in formulae and the like. We have referenced compounds as appropriate, as readers will expect us to, and feel that starting materials should not be treated any differently as to do otherwise would be subjective and non-systematic.

6. Following from the above, reference 34 describes a thorium-antimony complex. It's unclear what relevance this has in the present discussion of U-P species.

RESPONSE: The text in the manuscript associated with that reference is "for constructing heavier Group 15 motifs at actinide centres generally". Reference 34 details the synthesis of thorium-antimony bonds which we respectfully suggest fits the sentiment perfectly.

7. Again related to the above, the authors may consider clarifying some points by shortening sentences (the above example is a 7-liner).

RESPONSE: The above revisions have addressed this point naturally.

8. Page 3, line 7: the – after P should presumably be superscript.

RESPONSE: Since it is 'normal' text it is correct, but we realise would benefit from being prefixed by 'a', thank you for prompting us to check.

9. Figure 3: the caption describes parts (a) and (b), but only the species described in part (b) are shown. Not sure if the caption or the figure are incorrect here. Please change.

RESPONSE: We think the reviewer may be referring to figure 2 where they are correct there was an error which is corrected in response to reviewer 1 and with thanks to reviewer 2 also.

10. On the syntheses, the authors suggest the synthetic route is a promising pathway to access HPPH species, but they mention in almost all cases the multiple products form, which in some cases are inseparable (e.g. in the synthesis of 6). This, in my view, detracts from the utility. The authors might consider rephrasing such points.

RESPONSE: In the introduction we wrote "These insights offer potential to guide, and render more rational, future synthetic endeavours in this area generally." We feel that strikes the right tone.

11. The X-ray data for compound **8** are not of a publishable standard, with multiple peaks of residual density of between 2 and 4(!). The authors state that this is due to the heavy atom; if this was not fixed by an absorption correction, it is probably due to poor data quality (e.g. not a single crystal). This data should be recollected, and should not be published as it stands – such Q-peaks will greatly affect the accuracy of discussed metrical parameters in the core of the structure, especially given that these peaks are in the vicinity of both P and U.

RESPONSE: The X-ray data for compound **8** is of high quality, with an I/σ of 25.1 and R_{int} of 9.23% out to a resolution of 0.79 Å with good accuracy and precision of the bond lengths (to 3 or 4 dp and low σ values). We applied face indexed absorption corrections, but as the reviewer points out we still observe residual density peaks ranging from 2-4, which are each close to the U-P core. It is possible that these are an artefact of pseudo-merohedral twinning, with the observed monoclinic C cell having a β angle of 90.4°, close to 90°, which would give an apparent orthorhombic supercell. However, no twin law could be found that would account for this. In either case, the residual density of 4 is relatively small and not uncommon for structures containing a $Z = 92$ element (e.g. see *Nat. Commun.* **2023**, *14*, 4657) so the data are certainly publishable as they are.

Reviewer #3 (Remarks to the Author):

Review of “Isolable f-Element Diphosphene Complexes by Phosphinidene Group Transfer and Coupling at Uranium”.

Synopsis: The manuscript reports isolation of (HPPH)²⁻ stabilized by U(IV) centers, adding to f-element HNNH and HAsAsH complexes reported before. Combining experiments and detailed computational studies, the authors conclude reaction outcomes are controlled by the ancillary ligands, the radical nature of intermediates, and the phosphinidene substituent. They further propose the unfavourability of U adopting high (V/VI) oxidation states to form multiple polar covalent bonds to P.

Recommendation: Publish on a more specialized journal.

Since structurally relevant N, As, and P compounds have previously been reported, the synthetic novelty of the current study is limited. Probably the most important insight mentioned in this manuscript is “By contrast, the analogous one-electron oxidation of [UIV(TrenTIPS)(PH)]- results in the isolation of [UIV(TrenTIPS)(PH₂)] and II,49” Unfortunately, this is published in a previous work by the same group of authors. Overall, the authors have worked on this system for a long time and published a lot of works already. This work seems to be a mere extension of previous work. Therefore, it may suit a more specialized journal.

RESPONSE: We thank the reviewer for their thoughts and address their queries below. However, we respectfully disagree with the opinion that this is specialised work because: (i) molecular complexes containing HPPH remain exceedingly rare across the whole Periodic Table; (ii) these are the first f-element derivatives filling the gap created by reports of f-element complexes of HNNH and HAsAsH in 1992 and 2015, respectively; (iii) these HPPH linkages are made by a new methodology for HPPH; (iv) the work informs us about how these phosphinidene transfer reagents operate generally teasing out common underlying key reaction steps and then divergences and their reasons for occurring; (v) the work throws up the surprise result of the stability of open shell intermediates vs the anticipated electron paired formulations. It is true that we have worked with Tren^{TIPS} for a while now, but that is because it is so good at stabilising novel motifs, so whilst the ancillary ligand is usually the same the novel motif, and how it is made, is different each time so we believe it should not be confused as an extension but recognised for what it is, which is reporting a new f-element structural motif.

Major comments for improving the manuscript:

1. A Scheme showing relevant reactivity for As should be included in addition to Figure 1, so it will be easier for the readers to compare P and As.

RESPONSE: The reviewer makes a good suggestion, but figure 1 becomes quite congested and its value lies in its simplicity. Thus, to meet the reviewer’s sentiment we have amended the introduction to mention this since the reviewer is right to highlight that the HPPH here is constructed in an entirely different way to the acid-base chemistry that afforded analogous HNNH and HAsAsH linkages.

2. Reaction conditions (temperature, solvents, duration) and yields should be added in Figure 2.

RESPONSE: We understand the request but the scheme will become too cluttered. Also, for compounds that have an effective yield and an absolute yield due to sacrificial U-components providing a number in isolation is not helpful. However, all the details are provided in the experimental section in the manuscript.

3. No clear shoulder can be seen for compound 8 in the xT plot.

RESPONSE: The shoulder we refer to is in the χ vs T not χT vs T plot.

4. "The data are similar to those of $\{[(\text{TrenTIPS})\text{U}(\text{IV})]_2\{\mu\text{-h}_2\text{:h}_2\text{-(HAsAsH)}\}\}$ (MDCq U/As = 3.20/-1.12),²³ but overall are less polarized for 8 and certainly 9, suggesting that HPPH is a more effective donor to U than HAsAsH, or put another way HAsAsH is a better acceptor than HPPH, as predicted theoretically." This part is confusing. The charge on P is slightly more negative than that of As (-1.14 vs. -1.12). How the authors come to the conclusion that HAsAsH is better acceptor than HPPH? Same for the argument about the comparison between 8 and 9. The difference is subtle and purely DFT-based. The authors should restrain themselves from too many speculations.

RESPONSE: The reviewer is correct that the P charges are similar but we do not make our statement based on the P charges alone but also consideration of the U charges as well, where the U charge is much larger for the As complex than the P one, hence our comment "are less polarized". The same logic applies to the comparison of 8 and 9.

5. Some quantitative analysis should be done to verify the statement about the steric hindrance of different ancillary ligands. It is not obvious that three Ar*O ligands are less sterically demanding than Tren ligands. In addition, this reviewer doesn't understand why (P2)²⁻ can be isolated for TrenTIPS but not (HPPH)²⁻. The U-P distance should be longer for (HPPH)²⁻ than for (P2)²⁻, so the (HPPH)²⁻-bridging species should have very similar or slightly less.

RESPONSE: We believe the existing reaction profile calculations and associated discussion address these points already.

Minor comments:

1. For ref.20, it's a "Ge" complex, not "Ga". The typo showed up in page 2 and 8.

RESPONSE: We apologise for this error which has been corrected.

2. Reaction of P4 and molybdenum hydride precursors was firstly reported by Green and co-workers in 1974, while until 1977 the structure was confirmed by Cannillo et al., please add "J. Chem. Soc., Chem. Commun. 1974, 212." to ref.18.

RESPONSE: We have added the requested reference.

3. Figure 2 seems missing part "a) prior work showing the reaction of I with 1 to give the diphosphorus complex II and with 2 to give the phosphinidene complex III."

RESPONSE: The reviewer is correct, this was corrected in response to comments above with thanks.

4. There's a typo in ref.21, the page number should be "1081-1082" instead of "1981-1982".

RESPONSE: We apologise for this error which has been corrected.

Reviewer #4 (Remarks to the Author):

RESPONSE: We thank the reviewer for their contributions and their queries have been addressed above.

---End---

ROUND 2 COMMENTS

REVIEWER COMMENTS

Reviewer #1 (Remarks to the Author):

Liddle and co-workers have thoroughly addressed all of the reviewers' comments and clearly addressed the criticisms raised by the reviewer. Additionally, they have revised the manuscript in accordance with the reviewers' suggestions, which, in my view, has significantly improved the quality and readability of the manuscript.

I agree with the group's argument that this work is not merely an extension of their previous research, but rather fills an important gap in phosphorus and f-element chemistry through the successful formation of an HPPH^{2-} ligand. Therefore, I believe the work will strongly appeal to the readers of Nature Communications and is likely to inspire many chemists to further explore f-element chemistry using phosphinidene transfer reagents.

RESPONSE: We thank the reviewer for their constructive criticism which has improved the manuscript.